# The structural insight into the functional modulation of human anion exchanger 3

Liyan Jian [1,2,8], Qing Zhang[2,3,8], Deqiang Yao [2,4,8], Qian Wang[2], Moxin Chen[5,6], Ying Xia[2], Shaobai Li[2], Yafeng Shen[2], Mi Cao[2], An Qin [1,7] ✉, Lin Li[5,6] ✉ & Yu Cao [1,2] ✉

Anion exchanger 3 (AE3) is pivotal in regulating intracellular pH across excitable tissues, yet its structural intricacies and functional dynamics remain underexplored compared to other anion exchangers. This study unveils the structural insights into human AE3, including the cryo-electron microscopy structures for AE3 transmembrane domains (TMD) and a chimera combining AE3 N-terminal domain (NTD) with AE2 TMD (hAE3$^{NTD}$2$^{TMD}$). Our analyzes reveal a substrate binding site, an NTD-TMD interlock mechanism, and a preference for an outward-facing conformation. Unlike AE2, which has more robust acid-loading capabilities, AE3's structure, including a less stable inward-facing conformation due to missing key NTD-TMD interactions, contributes to its moderated pH-modulating activity and increased sensitivity to the inhibitor DIDS. These structural differences underline AE3's distinct functional roles in specific tissues and underscore the complex interplay between structural dynamics and functional specificity within the anion exchanger family, enhancing our understanding of the physiological and pathological roles of the anion exchanger family.

Cells meticulously regulate their intracellular pH within a narrow range to sustain an optimal environment for biochemical reactions and the functional integrity of organelles. Various channels and transporters are instrumental in cellular pH control, including proton pumps, sodium-potassium pumps, lactic acid transporters, and ammonium transporters[1–4]. To counteract the pH shifts caused by the accumulation of $CO_2$ and bicarbonate from energy metabolism and biosynthesis, bicarbonate transporters evolved to export the $HCO^{3-}$ anions, accompanied by either chloride influx ($Cl^-$/$HCO_3^-$ exchangers) or sodium influx ($Na^+$/$HCO_3^-$ cotransporter)[3,5]. Anion exchangers (AEs), belonging to the solute carrier superfamily SLC4, encompass three

members: AE1-3, encoded by the *SLC4A1-3* genes. AE1 predominantly acts as the $Cl^-$/$HCO_3^-$ exchanger in erythrocytes[6], while AE2 is crucial for pH homeostasis in acid-secreting cells such as osteoclasts and parietal cells[7,8]. Notably, AE3 is primarily expressed in excitable tissues, including the retina, heart, and brain[9–11], where it serves to counteract alkalosis and maintain cellular and organ functions. Disruption of AE3 function can lead to disturbances in pH homeostasis, potentially resulting in pathological conditions. In humans, loss-of-function mutations in AE3 have been linked with short QT syndrome, which may be caused by elevated intracellular pH when AE3 are deactivated[12,13]. Additionally, the Asp867 variant of AE3 is associated

[1]Department of Orthopaedics, Shanghai Key Laboratory of Orthopaedic Implant, Shanghai Ninth People's Hospital, Shanghai Jiao Tong University School of Medicine, Shanghai, China. [2]Institute of Precision Medicine, the Ninth People's Hospital, Shanghai Jiao Tong University School of Medicine, 115 Jinzun Road, Shanghai, China. [3]Structural Biology Program, Memorial Sloan Kettering Cancer Center, 1275 York Avenue, New York, NY, USA. [4]Institute of Aging & Tissue Regeneration, Renji Hospital, Shanghai Jiao Tong University School of Medicine, Shanghai, China. [5]Department of Ophthalmology, Shanghai Ninth People's Hospital, School of Medicine, Shanghai Jiao Tong University, Shanghai, China. [6]Shanghai Key Laboratory of Orbital Diseases and Ocular Oncology, Shanghai, China. [7]Department of Orthopaedics, Shanghai Frontiers Science Center of Degeneration and Regeneration in Skeletal System, Shanghai Ninth People's Hospital, Shanghai Jiao Tong University School of Medicine, Shanghai, China. [8]These authors contributed equally: Liyan Jian, Qing Zhang, Deqiang Yao. ✉e-mail: dr_qinan@163.com; lin_li@sjtu.edu.cn; yu.cao@shsmu.edu.cn

with common subtypes of idiopathic generalized epilepsy. Disruption of AE3 reduces the seizure threshold and decreases susceptibility to epileptic seizures[14–16]. Furthermore, animal models demonstrate that knockout or mutations of *Slc4a3* can induce retinal degeneration, progressing to retinitis pigmentosa[17,18], thereby positioning SLC4A3 as a potential candidate gene for human retinal diseases[19]. In summary, tissues with high metabolic rates, such as the heart and retina, rely on normal AE3 function to manage metabolic alkalosis and keep pH homeostasis.

As the anion transporter family, SLC4 genes could be categorized based on the mechanism driving carbonate/bicarbonate transport, three sodium-independent anion exchangers (AE1-3), five sodium-dependent co-transporters (NBCe1, NBCe2, NBCn1, NDCBE and NBCn2) and two unusual members (AE4 and BTR1)[3,20,21]. Within the SLC4 family, members exhibit high similarity in their transmembrane domains (TMDs) but display significant diversity in their N-terminal soluble domains (NTDs). For instance, human AE1-3 shares ~40–51% protein sequence identity in their NTDs, while the identities in their TMDs range from 67–71%. Furthermore, alternative mRNA splicing is a common occurrence across the NTDs of SLC4A members. A notable example is human AE1, which exists in two isoforms differing in NTD lengths: the kidney-type AE1 (kAE1) lacks the first 65 residues found in the erythrocyte-type AE1 (eAE1)[6,22]. In the case of human AE3, two major forms have been identified cardiac AE3 (cAE3) and brain AE3 (bAE3), with cAE3 missing residues 1-296 present in bAE3[3,23,24]. Previous structural studies on AE1 and AE2 have elucidated their inward- and outward-facing conformations, as well as the substrate-binding pockets in these different states[25–28]. Additionally, the full-length structures of human AE2 revealed TMD-NTD interplay mediating the conformational change during the anion exchange[27]. As one of the major pH and carbonate mediators in excitable tissues, AE3 demonstrates exchange activities that are distinctly regulated compared to AE1 and AE2. While its deficiency could cause a severe pH disorder in the heart and brain, intriguingly, AE3's transport activity is about 60–70% lower than that of AE1 and AE2 in recombinant expression systems[29,30]. Unlike AE2, which is highly pH-sensitive, AE3 shows relative pH-independence, functioning stably across a range of pH levels[29]. Moreover, AE3 is more sensitive to the anion transporter inhibitor DIDS (4,4′-diisothiocyana-tostilbene-2,2′-disulfonate) than AE2, with an IC50 of about 300-fold less than that of AE2[30]. These functional differences within the AE subfamily suggest subtle yet critical structural variations in AE1-3, underpinning their specialized adaptations to meet the distinct demands of various cells and tissues.

Here, we determined the cryo-EM structures of human AE3 in its TMDs-only form and its soluble domains in an AE2-AE3 chimera protein. These structures have unveiled the local arrangement of the HCO3−binding pocket, the NTD-TMD interlock, as well as the specific interactions between AE3 and the anion transporter inhibitor DIDS. Combining the structural and functional studies, our findings provide insights into the molecular basis of AE3's activity and its regulation, particularly in the context of inhibitor binding and pH stability.

## Results

### The cryo-EM structural determination of human AE3 (hAE3)
To investigate the role of AE3 in pH mediation precisely, we utilized the HEK293F cell line, which was genetically engineered to eliminate the endogenous *SLC4A2* gene. This gene produces AE2, the primary anion exchanger in these cells (Supplementary Fig. 1a). The successful knockdown of AE2 expression was confirmed through immunoblot analysis (Supplementary Fig. 1b). This targeted genetic modification allowed for a more isolated examination of heterogeneously expressed AE1-3's functionality in regulating cellular pH, minimizing the confounding effects of endogenous AE2. Subsequent overexpression of AE3 in the HEK293F *SLC4A2*-KO cell line revealed a lower pH-modulating function than that observed for AE2 (Fig. 1a, b). It was also

noted that the activities of AE2 and AE3 could be inhibited by adding DIDS, a broad-spectrum anion transporter inhibitor (Fig. 1a, b and Supplementary Fig. 13). Various splicing isoforms and expression constructs of AE3 were evaluated for their expression levels and thermostabilities during purification processes. The cardiac isoform hAE3 (the amino acid 300-1232 as the numbering of UniProt ID P48751-1, [available at https://www.uniprot.org/uniprotkb/P48751/entry#sequences]) demonstrated the highest stability and yield, as assessed by size-exclusion chromatography (Supplementary Fig. 1c) and negative-stained EM analysis. The hAE3 proteins were solubilized from the cell membrane using the detergent lauryl maltose neopentyl glycol (LMNG) and subsequently purified with size-exclusion chromatography (SEC), employing glyco-diosgenin (GDN) as the detergent in the mobile phase. Like other SLC4A family members reported in previous studies[25,27,31,32], hAE3 maintained stable homodimer formation throughout the purification process, as confirmed by SEC and 200 kV cryo-EM analysis. High-resolution cryo-EM images of hAE3 were captured using a Titan Krios transmission electron microscope (FEI) operated at 300 kV. The data processing with RELION-3 and cryoSPARC[33,34]. The structure of hAE3 in the presence of HCO3− and HCO3−/DIDS, and without HCO3− supplementation were determined with C2 symmetry, reaching overall resolutions of 2.7 Å, 2.7 Å, and 2.9 Å, respectively (Supplementary Fig. 2–4). The structural models were constructed within the corresponding EM maps, using the cryo-EM structure of human AE2 as the starting model, which represents an outward-facing conformation with HCO3− bound (PDB ID 8GVC). Given the minimal backbone differences among these three structures (overall RMSD about 0.19 - 0.21 Å), the hAE3 structure supplemented with HCO3− (hAE3HCO3−) is described here as the representative model.

### The overall structure of human AE3
In general, human AE3 adopts a homodimeric state, and each protomer is composed of a transmembrane domain (TMD) and an N-terminal cytoplasmic domain (NTD), the latter remaining unresolved in all three cryo-EM structures (Fig. 1c). In TMDs, the cryo-EM density map was fragmentary for residues 860–883 in the extracellular loop at the dimeric interface and twenty-two residues of the C-terminal loop, and thus left unmodeled (Fig. 1d, e). In the human AE2 structure, a prominent loop connecting TMH10/11 extends from the TMD to the soluble NTD, playing a role in TMD-NTD interactions[27]. Although the NTD of AE3 remains unsolved, the loop between TMH10 and 11 in the hAE3 structure exhibits a similar arrangement and local structure to that of AE2 (Figs. 1d, 2b). Moreover, the PGDKP sequence in the loopTMH10/11, a critical region in NTD-TMD interactions of AE2, is conserved in AE2 (residues 1077–1081) and AE3 (residues 1069–1073), suggesting a potential analogous interaction between the NTD and TMD in AE3. Similar to AE1 and AE2, the TMD of hAE3 comprises fourteen transmembrane helices, TMH1-14, seven short intracellular helices IH1-7, and one extracellular helix EH1 (Fig. 1f). These helices are spatially organized into two distinct subdomains within the membrane: the 'core' subdomain, formed by TMH1-4 and TMH8-11, and the 'gate' domain, comprised of TMH5-7 and TMH12-14 (Fig. 1g). The gate and core subdomains are connected by extracellular loops bridging TMH11/12 and TMH7/8 and an intracellular loop linking TMH4/5, along with helices EH1 and IH3. From an extracellular perspective, a deep 'cleft' is evident between the gate and core subdomains, housing the non-proteinaceous density indicative of the substrate binding site (Fig. 1g).

In the AE3 TMD structures, the dimerization of hAE3 is primarily facilitated by hydrophobic interactions between the transmembrane helices TMH6 of the two protomers (Fig. 2a). Key residues oppositely arranged at the dimeric interface, including L894, I898, F905, and F909, form hydrophobic pairs that contribute significantly to the stability of the dimer. In addition to these interactions, the extracellular loop connecting TMH5 and TMH6 and the intracellular loop between

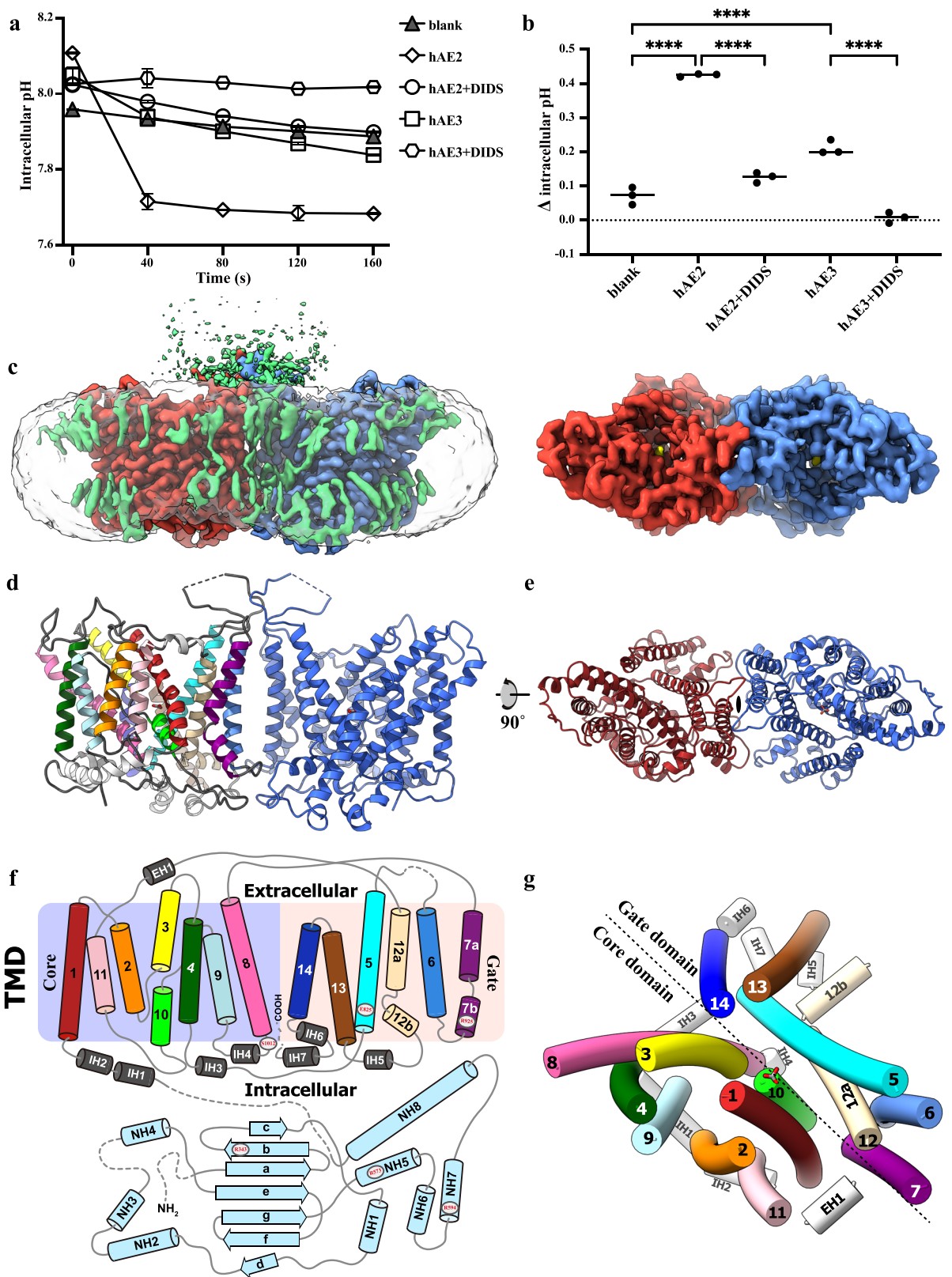

TMH6 and TMH7 also play a role in hAE3 dimerization. This stabilization is achieved through hydrogen bonds formed between the side chains of N891-N891 and the backbone carbonyl and imino group of Y856-Y856 (Fig. 2c). Moreover, interactions involving R924-H1140, R917-H1140, and L919-L1135 further contribute to the dimerization of hAE3(Fig. 2d). Notably, the contacts between the TMDs of hAE3 protomers are relatively sparse. Based on structural insights from the AE1

and AE2, we postulate that the unresolved N-terminal domains (NTDs) of AE3 might also play a crucial role in dimerization.

## The anion-binding pocket

In all three cryo-EM structures, hAE3 TMD consistently adopts an outward-facing conformation, demonstrating a high degree of structural conservation compared to human AE1 and AE2 in their respective

**Fig. 1 | The overall structure and the arrangement of transmembrane helices of human AE3. a** and **b** Functional comparison of AE2 and AE3 in regulating intracellular pH. Changes in intracellular pH upon Cl⁻ reintroduction were monitored in *SLC4A2*-KO HEK293F cells overexpressing the human *SLC4A2* or *SLC4A3* genes, both in the presence and absence of the DIDS inhibitor (40 μM). Cells without overexpression served as controls. All experiments replicated independently three times to ensure consistency. One-way ANOVA with Tukey's multiple comparisons test was performed, ****$p < 0.0001$. Source data are provided as a Source Data file. **c** The cryo-EM density map of hAE3$^{HCO_3^-}$ viewed parallel to the cell membrane (left) and from the extracellular perspective (right). The cryo-EM density map corresponding to two protomers in dimeric hAE3$^{HCO_3^-}$ were highlighted in red (chain A)

and blue (chain B), respectively. Non-protein densities within the TMDs are depicted in gray, with lipid molecules and unresolved protein segments in green. **d** The dimeric architecture of hAE3$^{HCO_3^-}$ TMD aligned with the orientation of the EM map above. The protomer B was colored in blue and protomer A was colored by helices in a rainbow gradient. **e** The cartoon representation of dimeric structure of hAE3$^{HCO_3^-}$ TMD viewed from the same perspective as the EM map. **f** Topological diagram of the hAE3 monomer, with TM helices color-matched to those in Fig. 1d. Disease-associated mutation sites are indicated with circles and red font. **g** Arrangement of transmembrane helices in hAE3, viewed from the extracellular side, illustrating the spatial organization of the helices.

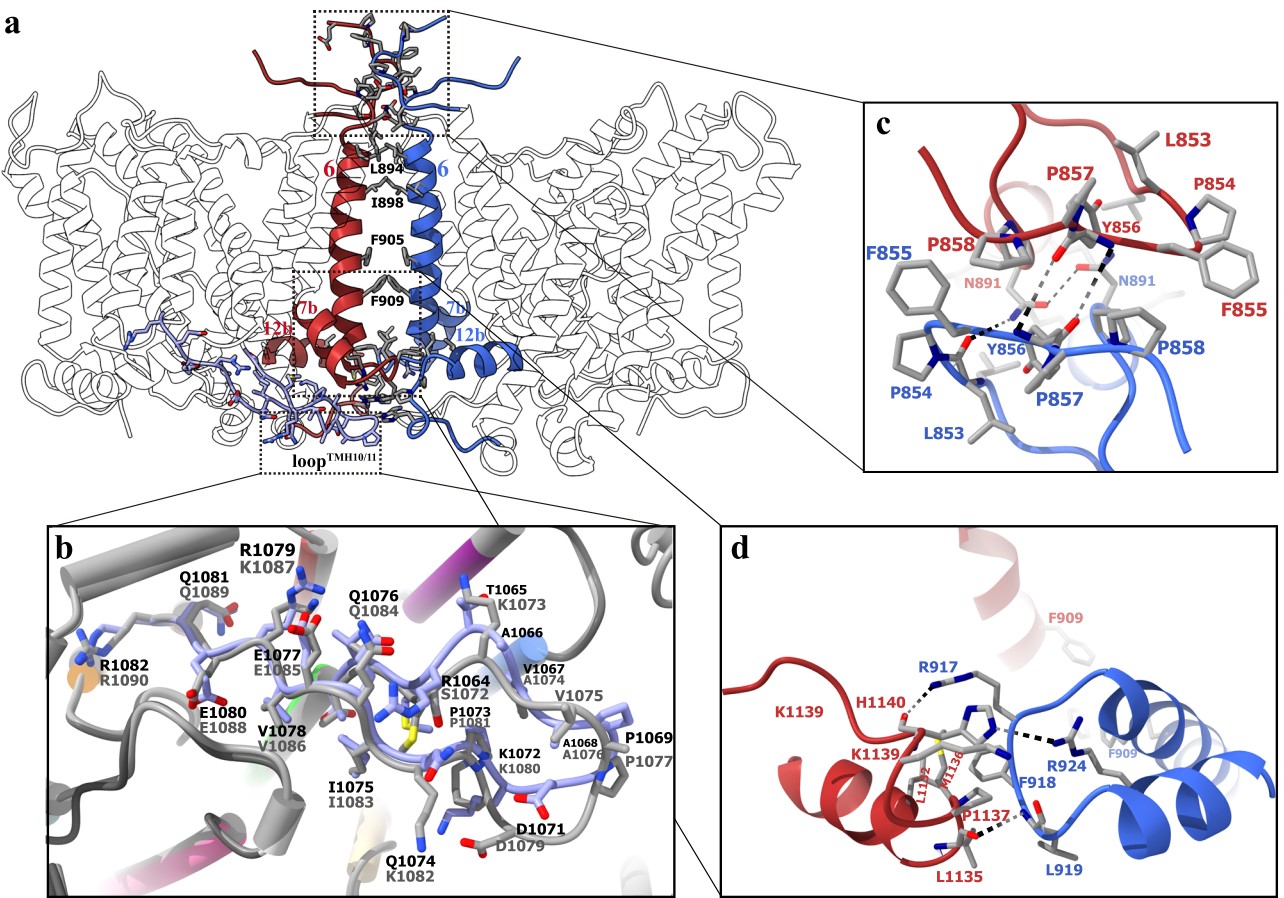

**Fig. 2 | The dimeric interface and the loop$^{TMH10/11}$ of hAE3. a** The hAE3 dimer is depicted in a cartoon representation, with dimerization-contributing helices and loops highlighted in red (protomer A) and blue (protomer B). The contacting residues in the dimeric interface were shown as stick model colored by elements. The loop$^{TMH10/11}$ of protomer A was shown in a combined stick and cartoon model, colored in slate blue for clarity. **b** The comparative analysis of the loop$^{TMH10/11}$ from AE2 and AE3. The TMDs of hAE2 and hAE3 were structurally superposed and their

loop$^{TMH10/11}$ were shown as stick and cartoon model colored in slate blue and gray for hAE3 and hAE2, respectively. Residues are labeled for identification, with hAE3 labels in black and hAE2 labels in gray. **c** The dimeric interactions of AE3 at the extracellular side. **d** The dimeric interactions of AE3 at the cytoplasmic side. The interacting residues were shown as stick model and colored by elements, highlighting the molecular contacts facilitating dimerization.

outward-facing conformations (Supplementary Figs. 5a, b)[25,27]. The TMD of hAE3 forms a deep outer vestibule, as determined with HOLLOW[35], composed of tiled transmembrane helices TMH1, 3, 5, 8, 13, and 14, with TMH10 sealing its bottom (Fig. 3a, b). Surface potential analysis of the outer vestibule indicates an interior surface that is predominantly hydrophobic or negatively charged near the exit, transitioning to positively charged at the bottom. This area includes three spots corresponding to surface-exposed basic residues K842/845, K1173, and R1052 (Supplementary Fig. 5e). Notably, this charge configuration is conserved among anion exchangers; AE1 exhibits a similar pattern to AE3, while AE2 differs by lacking a positively charged spot corresponding to K1173 in AE3, resulting in a more negatively

charged outer vestibule (Supplementary Fig. 5c–e). Within hAE3's outer vestibule, a non-proteinaceous density was identified, with the HCO₃⁻ anion modeled therein. At the bottommost part of the vestibule, R1052 from helix TMH10, analogous to R730 in AE1 and R1060 in AE2, is positioned near the HCO₃⁻ and stabilizes it through electrostatic interaction (Fig. 3a, c). The R1052A mutation in hAE3 significantly impairs its pH-modulating ability, underscoring its crucial role in substrate binding (Fig. 3d).

DIDS, a pan-inhibitor affecting anion transport in the SLC4, SLC16, and SLC26 families[3,36–38], forms covalent bonds with the ε-amino group(s) of lysine residues in AE1 and AE2, contingent on lysine availability and reaction pHs[27,39]. In the hAE3-DIDS structure, the cryo-EM

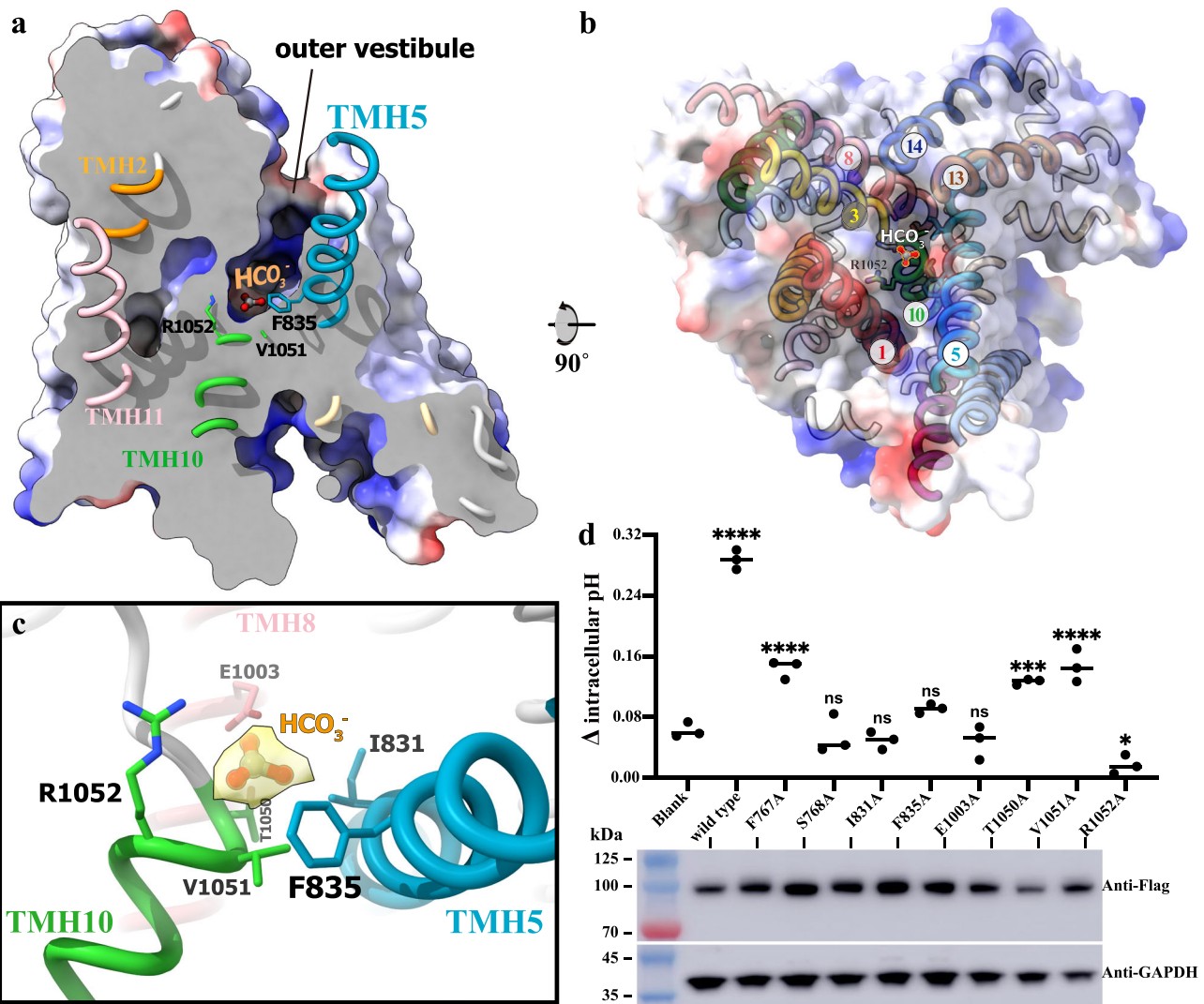

**Fig. 3 | The substrate binding site in the outer vestibule of hAE3. a** The outer vestibule and the substrate-binding site in hAE3 TMD. The hAE3$^{HCO_3^-}$ was shown as solvent-accessible electrostatic surface–potential maps, cut away to reveal the inner details of the outer vestibule. The helics TMH5,10–12 were shown as cartoon model colored to the same scheme as shown in Fig. 1a. The $HCO_3^-$ anion and the key surrounding residues were shown as stick model and colored by elements. **b** Helical arrangement within the outer vestibule. The hAE3$^{HCO_3^-}$ was shown as the solvent-accessible electrostatic surface–potential maps and rendered translucent to facilitate observation of the helix organization. Essential helices encircling the vestibule are identified by their TMH numbers. **c** The interactions among the $HCO_3^-$ and hAE3. The $HCO_3^-$ anion were shown as stick-ball model, and the interacting residues were shown as stick model, all being colored by elements. The cryo-EM density map corresponding to the $HCO_3^-$ was shown in yellow. **d** Functional impact of outer vestibule surface residues on AE3's pH regulation. Upper panel: the pH-mediating functions of hAE3 wild type and variants with mutations on the surface of outer vestibule, with all experiments replicated independently three times to ensure consistency. Lower panel: Representative immunoblot results of cell lysates expressing wild-type hAE3 and the corresponding mutants, showcasing protein expression levels. One-way ANOVA with Tukey's multiple comparisons test was performed, ns: $p > 0.05$, *: $p < 0.05$, ***: $p < 0.001$, ****: $p < 0.0001$. Source data are provided as a Source Data file.

density map of DIDS extends continuously to the sidechain of K842, indicating a covalent linking (Fig. 4a, b, and Supplementary Fig. 6). Viewed from the extracellular side, DIDS is positioned near the outer vestibule's exit/entrance atop the $HCO_3^-$ anion, effectively obstructing the pathway between the substrate-binding site and the extracellular environment. Structural analysis of DIDS-contacting residues reveals that the core stilbene framework is encircled by hydrophobic residues such as L772/729, F726/835/1181, V773, and I831, which facilitate hydrophobic interactions stabilizing DIDS (Fig. 4c, d). Additionally, polar residues like T725, S1178, and E838 are near the stilbene's isothiocyano groups. Interestingly, the residue K1173, a potential reactive site, appears unbonded with DIDS, as indicated by its separate density. This observation aligns with previous findings that lower pH levels hinder the linkage between K851 in AE1 (the structural analog of K1173) and DIDS[39].

## Full-length structure of NTD$^{AE3}$-TMD$^{AE2}$ chimera

Despite extensive efforts, the determination of the full-length structure of human AE3 was unsuccessful, with the soluble N-terminal domain (NTD) remaining unresolved. This challenge is likely attributed to the highly dynamic and flexible nature of the AE3 NTD in its outward-facing state. Attempts to capture AE3 in its inward-facing state, similar to the methods used in resolving the full-length structure of AE2[27], including varying pHs and anions in cryo-EM sample preparation, did not yield alternative conformations of AE3. To overcome this obstacle and gain structural insight into the AE3 NTD, we constructed a chimeric anion exchanger, combining the NTD of hAE3 (297–671) with the TMD of hAE2 (672–1241). This chimera, termed hAE3$^{NTD}$2$^{TMD}$, was designed to exhibit the inward-facing conformation with a visible NTD, capitalizing on the thermostability of human AE2's inward-facing state. The hAE3$^{NTD}$2$^{TMD}$ chimera exhibited excellent stability and homogeneity as assessed

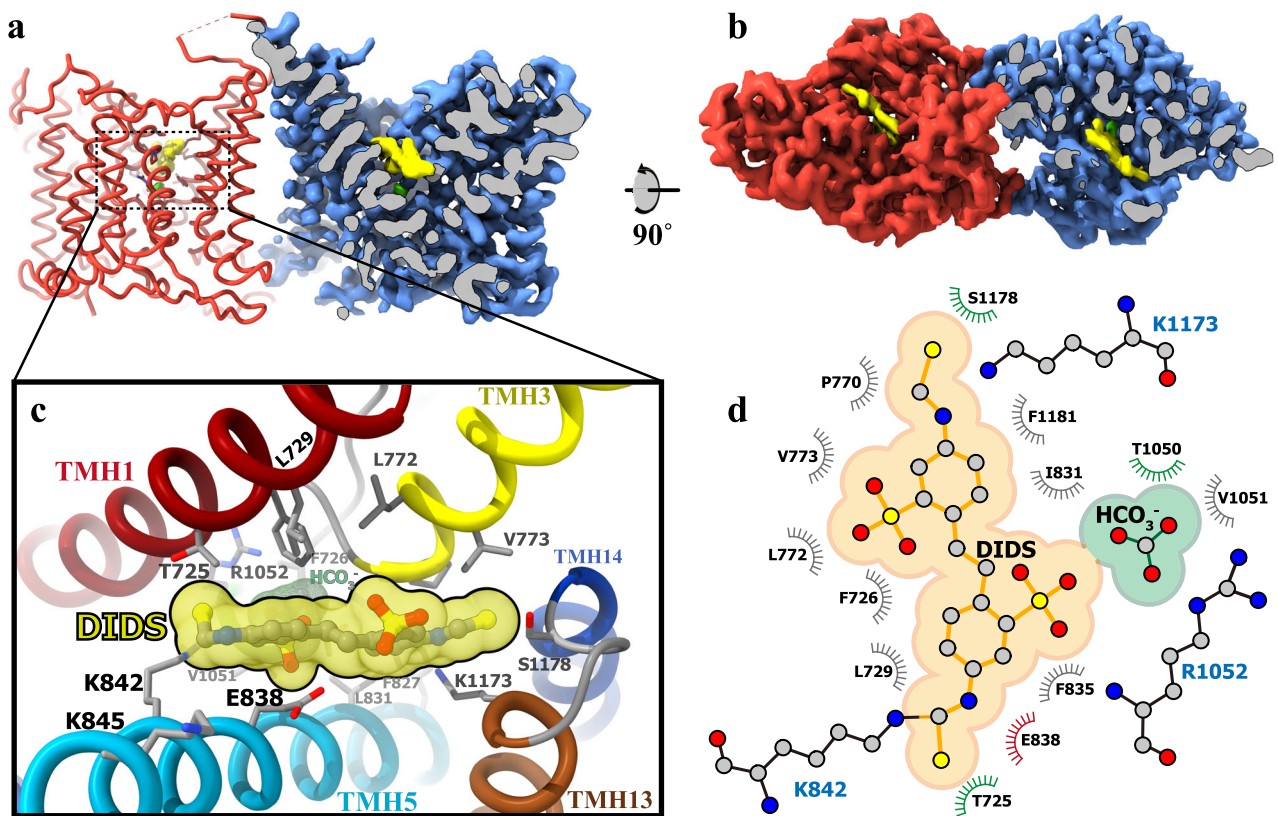

**Fig. 4 | DIDS Inhibitor Binding in hAE3's Outer Vestibule. a** The cryo-EM map of hAE3$^{HCO_3^-/DIDS}$ as viewed parallel to the cell membrane. The non-proteinous densities corresponding to HCO$_3^-$ and DIDS were colored in green and yellow, respectively. For clarity, the cryo-EM density map of protomer B (blue) is sectioned to reveal these densities within the outer vestibule. The density map corresponding to protomer A were omitted to display the cartoon model of hAE3$^{HCO_3^-/DIDS}$. **b** The cryo-EM map of the hAE3$^{HCO_3^-/DIDS}$ as viewed from the extracellular side. **c** Close-

up of DIDS interactions with hAE3. The hAE3$^{HCO_3^-/DIDS}$ was shown as cartoon model, focusing on the interaction site. The DIDS was shown as stick-ball model colored by elements and solvent-accessible surface model in yellow. Surrounding residues engaging with DIDS were shown as stick model and colored by elements. **d** The schematic diagram for the hAE3s-DIDS interaction network, generated using LigPlot+ (v2.2.4), elucidating the molecular contacts between DIDS and hAE3.

through size-exclusion chromatography (Supplementary Fig. 1d). Functional assays in HEK293 cells confirmed the pH-mediating activities of the hAE3$^{NTD}$2$^{TMD}$ chimera, and both 2D and 3D-classifications displayed EM signals indicative of a full-length anion exchanger (Fig. 5a, b). The cryo-EM structure of hAE3$^{NTD}$2$^{TMD}$ was determined at an overall resolution of 3.35 Å, with both AE3 NTD and AE2 TMD clearly resolved in the EM map (Supplementary Fig. 7).

In the hAE3$^{NTD}$2$^{TMD}$ structure, similar to the full-length human AE2 structure, a homodimer formation is observed where the TMDs adopt an inward-facing conformation (Fig. 5c, d). The structural superposition of hAE2 and hAE3$^{NTD}$2$^{TMD}$ revealed high similarity, with an overall RMSD (root-mean-square deviation) of 1.813 Å (Supplementary Fig. 8). The AE3 NTD comprises seven β-strands (a-g) arranged in a planar configuration, sandwiched by eight helices (NH1-8) (Fig. 1f). Notably, helix NH8, the longest in AE3, penetrates deeply into the adjacent NTD in the dimeric structure, extending from its N-terminal end towards the TMD at its C-terminal end (Fig. 5c, d). Dimerization of the AE3 NTDs is primarily facilitated by interactions among the loops linking NH7/8 (loop$^{NH7/8}$) and NH1/β strand c (loop$^{NH1/c}$), as well as helix NH8 from both protomers. The primary dimeric interface of the NTD (interNTD interface 1) is situated near the C2 symmetry center (Fig. 5f), where loop$^{NH7/8}$ and loop$^{NH1/c}$ are arranged in an alternating, antiparallel pattern. A network of hydrogen bonds is formed between oppositely-situated residue pairs, including S608-I611, I609-I611, L366-V610, and F368-S608. Additionally, the amine side chain of Q629 forms hydrogen bonds with S614 and P612 from the neighboring NTD, reinforcing the hydrogen bond network (Fig. 5f). Flanking the C2 symmetry center are

two smaller dimeric interfaces (interNTD interface 2), characterized by electrostatic interactions between E615-R637 and K368-E617 and a hydrogen bond between P361 and D620 (Fig. 5g).

Previously reported human AE2 structure revealed an NTD-TMD interlock, characterized by interactions between its NTD and the loop connecting transmembrane helices 10 and 11 (loop$^{TMH10/11}$), which stabilizes the AE2 TMD in the inward-facing conformation[27]. In the hAE3$^{NTD}$2$^{TMD}$ structure, the AE3 NTD engages with the AE2 TMD in a manner reminiscent of AE2's own NTD-TMD interactions, with the loop$^{TMH10/11}$ extending from the TMD into the NTD surface. However, a detailed structural comparison reveals distinct differences in the interacting residues between hAE3$^{NTD}$2$^{TMD}$ and hAE2. Despite the high conservation of the motif sequence AV(I$^{AE3}$)APGDKP, variations in the amino acid of loop$^{NH7/8}$ of NTDs lead to altered interaction dynamics (Supplementary Fig. 8c). Specifically, substituting Q618$^{AE2}$ with E617$^{AE3}$ disrupts the original contact between Q618$^{AE2}$ and D1079$^{AE2}$ in AE2. This results in a new electrostatic interaction between E617$^{AE3}$ and K1080$^{AE2}$ in the hAE3$^{NTD}$2$^{TMD}$ chimera (Fig. 5e, Supplementary Figs. 8a, b). This alternative interaction draws the backbone of AE3 loop$^{NH7/8}$ slightly closer to the loop$^{TMH10/11}$ in the hAE3$^{NTD}$2$^{TMD}$ chimera than its positioning in AE2. As a result, the position of E621$^{AE2}$ was occupied by L621$^{AE3}$, which eliminates another pivotal hydrogen bond present in the AE2 NTD-TMD interlock, specifically between E621$^{AE2}$-G1078$^{AE2}$ (Supplementary Fig. 8b). While the full-length structure of AE3 remains unresolved in this study, the structural insights gained from the hAE3$^{NTD}$2$^{TMD}$ chimera structure, coupled with the conserved sequence of loop$^{TMH10/11}$ sequence, suggest that the NTDs of AE2 and AE3 engage with their respective TMDs in

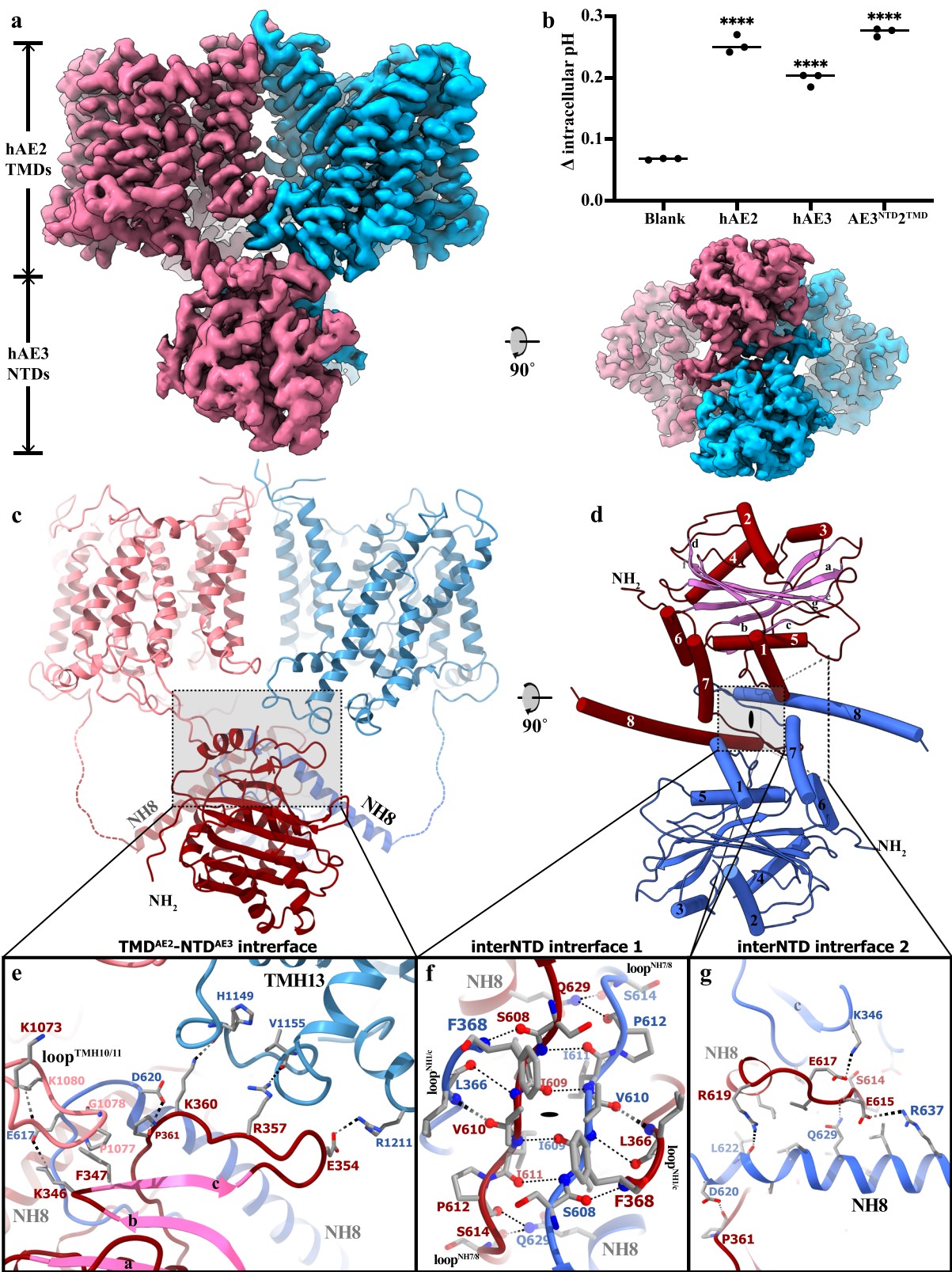

distinct interaction patterns. These differences likely contribute to the varied pH responses observed in their regulatory functions.

## The mechanistic implication for anion exchanging by human AE3

Capturing the inward-facing conformation of AE3 has proven challenging. However, the considerable sequence identity (~71%) between the TMDs of AE2 and AE3 renders the hAE3$^{NTD}$2$^{TMD}$ chimera structure a valuable template for homology modeling. Employing the SWISS-MODEL homology modeling pipeline (https://swissmodel.expasy.org/ interactive), we generated the hAE3 TMD structure in the inward-facing state with a QMEANDisCo Global score of 0.72, indicating a reliable model prediction[40]. The modeled hAE3 TMD exhibits a cytoplasm-open conformation akin to that of hAE2, as evidenced in

**Fig. 5 | The structural analysis of AE3$^{NTD}$2$^{TMD}$ chimera. a** The cryo-EM map for AE3$^{NTD}$2$^{TMD}$ chimera as viewed parallel to the cell membrane (left) and from the cytoplasmic perspective (right). The cryo-EM density map corresponding to two protomers in dimeric AE3$^{NTD}$2$^{TMD}$ chimera were shown in magenta (chain A) and cyan (chain B), respectively. **b** Functional assays of the AE3$^{NTD}$2$^{TMD}$ chimera conducted in cell-based systems, with all experiments replicated independently three times to ensure consistency. One-way ANOVA with Tukey's multiple comparisons test was performed, ****$p < 0.0001$. Source data are provided as a Source Data file. **c** The structure of AE3$^{NTD}$2$^{TMD}$ chimera viewed from the same viewpoint as the EM map shown above. The protomer A of AE3$^{NTD}$2$^{TMD}$ chimera was colored in magenta and red for its TMD and NTD, respectively. The protomer B of AE3$^{NTD}$2$^{TMD}$ chimera was colored in cyan and blue for its TMD and NTD, respectively. **d** The structure of dimeric AE3 NTD was shown as cartoon model, detailing helices NH1-8 and β-strands a–g. **e** Zoomed-in view of the interactions between AE3 NTD and AE2 TMD within AE3$^{NTD}$2$^{TMD}$ chimera, focusing on the interface detailed in (**c**) by the dotted frame. The contacting residues in AE2$^{TMD}$-AE3$^{NTD}$ interface were shown as stick model colored by elements. **f, g** Detailed examination of the interactions mediating the dimerization of AE3 NTDs. The dotted frames in Fig. 5d were enlarged, and the contacting residues in dimeric interface of AE3 NTD were shown as stick model colored by elements. The essential backbone carbonyl and imino groups contributing to hydrogen bonds formation in interNTD interface 1 were represented as spheres.

Supplementary Figs. 9a, b. Notably, while the gate domain retains its helical configuration, the core domain helices, particularly TMH1, 3, and 10, demonstrate significant tilting and translational shifts between the inward- and outward-facing states (Fig. 6a). Consequently, the outer vestibule constricts into a diminutive pocket in the inward-facing conformation of hAE3, concurrently giving rise to an inner vestibule on the cytoplasmic side of the TMD. Furthermore, the crucial HCO$_3^-$-interacting residue, R1052, transitions from its position at the outer vestibule's base to the bottom of the emergent inner vestibule, priming it for substrate uptake from the cytosol (Fig. 6b).

Our EM analysis suggested that AE3 predominantly adopts an outward-facing conformation in its resting state, contrasting with AE2's inward-facing resting state[27]. While the full-length AE3 structure in its outward-facing state remains to be delineated, insights from the hAE3$^{NTD}$2$^{TMD}$ chimera hint at a similar NTD-TMD interlock mechanism facilitating AE3's conformational transitions, akin to those observed in AE2[27]. The NTD is poised to interlock with the TMD in response to intracellular pH shifts or substrate changes, with the core domain's helical rearrangements unveiling the inner vestibule for substrate engagement. The inward-facing state of AE3 is suggested to be relatively unstable and transient, as inferred from the dominance of the outward-facing conformation in EM observations. This implies that the substrate-laden, inward-facing AE3 may rapidly revert to its outward-facing conformation to release the substrate, thereby completing a functional cycle.

## Discussion

AE1-3 constitute a family of anion exchangers that play pivotal roles in modulating intracellular pH and bicarbonate levels. Despite their similar functions, variations in tissue distribution suggest distinct cellular roles for each. AE2, for instance, exhibits a pronounced acid-loading effect in response to significant alkalosis challenges in acid-secreting tissues like osteoclasts and parietal cells. AE3, on the other hand, is primarily found in excitable tissues like the retina, heart, and brain. Maintaining pH homeostasis is especially important in these environments, as electrical activity can induce rapid pH changes. These acid-base fluctuations may impact physiological processes by affecting various ion channels[41,42]. Thus, ensuring gradual and moderate pH changes in excitable tissues is crucial. Compared to AE2, AE3's pH-modulating activity is lower and exhibits less pH sensitivity[29]. The cryo-EM structures of AE3 showed an NTD-TMD interlock in a similar configuration but with subtle differences compared with AE2. Key electrostatic interactions and hydrogen bonds integral to AE2's interlock, specifically E508-K1073, Q618-D1079, and E621-G1078, are absent in AE3 due to amino acid variations within the loop$^{TMH10/11}$ and loop$^{NH7/8}$ (Supplementary Fig. 8). These alterations likely contribute to the instability of AE3's inward-facing conformation, potentially diminishing its efficiency in bicarbonate capture and anion exchange compared to AE2. The sequence alignment between AE1 and AE2 reveals that the interactions Q618-D1079 and E621-G1078 are also absent in AE1 (Supplementary Fig. 8c). As a result, AE1 also predominantly adopts an outward-facing conformation in previous structural studies[25,26,43,44]. In addition to the NTD-TMD interlock, another self-regulation

mechanism identified in AE2 is its inner vestibule occupancy by the C-terminal loop[27]. However, lacking the inward-facing conformation in our cryo-EM structure makes the self-inhibitory effect uncertain in AE3.

Another distinguishing feature of AE3 is its heightened sensitivity to the pan-anion exchanger inhibitor DIDS relative to AE2[30]. The increased vulnerability may stem from the predominant outward-facing conformation of AE3 under various tested pH and anion conditions, which renders its DIDS-binding site more accessible from the extracellular milieu. In contrast, AE2's inward-facing resting state may impede DIDS access to its binding site, rendering it less susceptible to inhibition.

In previously documented AE family structures containing DIDS or H$_2$DIDS, such as the crystal structure of AE1-DIDS (PDB ID 4YZF), the cryo-EM structures of AE1-DIDS (PDB ID 8T6V), AE1-H$_2$DIDS (PDB ID 7TY6), and AE2-DIDS (PDB ID 8GV8)[25,27,39], an empty substrate-binding site is typically observed. The inhibitory effect of DIDS/H$_2$DIDS on substrate association in AE1 or 2 might be attributed to charge repulsion, as indicated by the proximity−3.8 Å and 4.7 Å−between the sulfonic acid groups of DIDS and the bicarbonate-binding sites in AE1 and AE2, respectively (Supplementary Fig. 10). In contrast, in the hAE3$^{HCO3−/DIDS}$ structure, the distance between the sulfonic acid groups of DIDS and the bicarbonate ion exceeds 5.4 Å, allowing for their simultaneous presence within the outer vestibule. This structural arrangement underscores the unique interaction dynamics within AE3 compared to its counterparts, contributing to its distinct functional and inhibitory response profiles.

Phosphatidylinositol 4,5-bisphosphate (PIP$_2$) has been found in the structures of human AE1, AE2, and BTR1[28,43,45], which were previously reported as activators of NBCe1 and NBCn1[46,47]. Mutations in the PIP$_2$ binding site in AE2 and BTR1 led to reduced substrate transport activity, indicating the important mediating function of PIP$_2$ for SLC4 proteins[28,45]. In our structure of the NTD$^{AE3}$-TMD$^{AE2}$ chimera, densities of PIP$_2$, cholesterol, and CHS in the TMD of AE2 (Supplementary Fig. 11a) correspond with the previous AE1 and AE2 models[27,28,43], further confirming the lipid regulation of human AE2. While the cryo-EM maps of AE3 did not provide sufficient density to precisely model PIP2 or other lipids at the corresponding PIP2 binding site of AE2, we did observe non-proteinous density indicative of a potential lipid molecule (Supplementary Fig. 11b–d). Further structural analysis revealed that key inositol phosphate-interacting residues in AE2, such as R925/932/933, K1147, and H1148, are highly conserved in AE3 (Supplementary Figs. 11e, f). This conservation suggests that AE3 might similarly bind an acidic lipid, potentially playing a role analogous to PIP$_2$ in AE2 modulation. Future experiments, including mutating these residues in AE3, are warranted to gain deeper insights into the regulatory mechanisms of AE3.

Several pathogenic mutations in AE3 have been associated with epilepsy and short QT syndrome. One notable mutation site is R925[13], located on the cytoplasmic side of the TMDs near the potential binding pocket for the hydrophilic head of phospholipid, such as the inositol phosphate moiety of PIP$_2$ (Supplementary Fig. 12). Although our previous studies did not confirm the identity of the phospholipid in the AE3 TMD, the proximity of R925 to this region suggests its potential

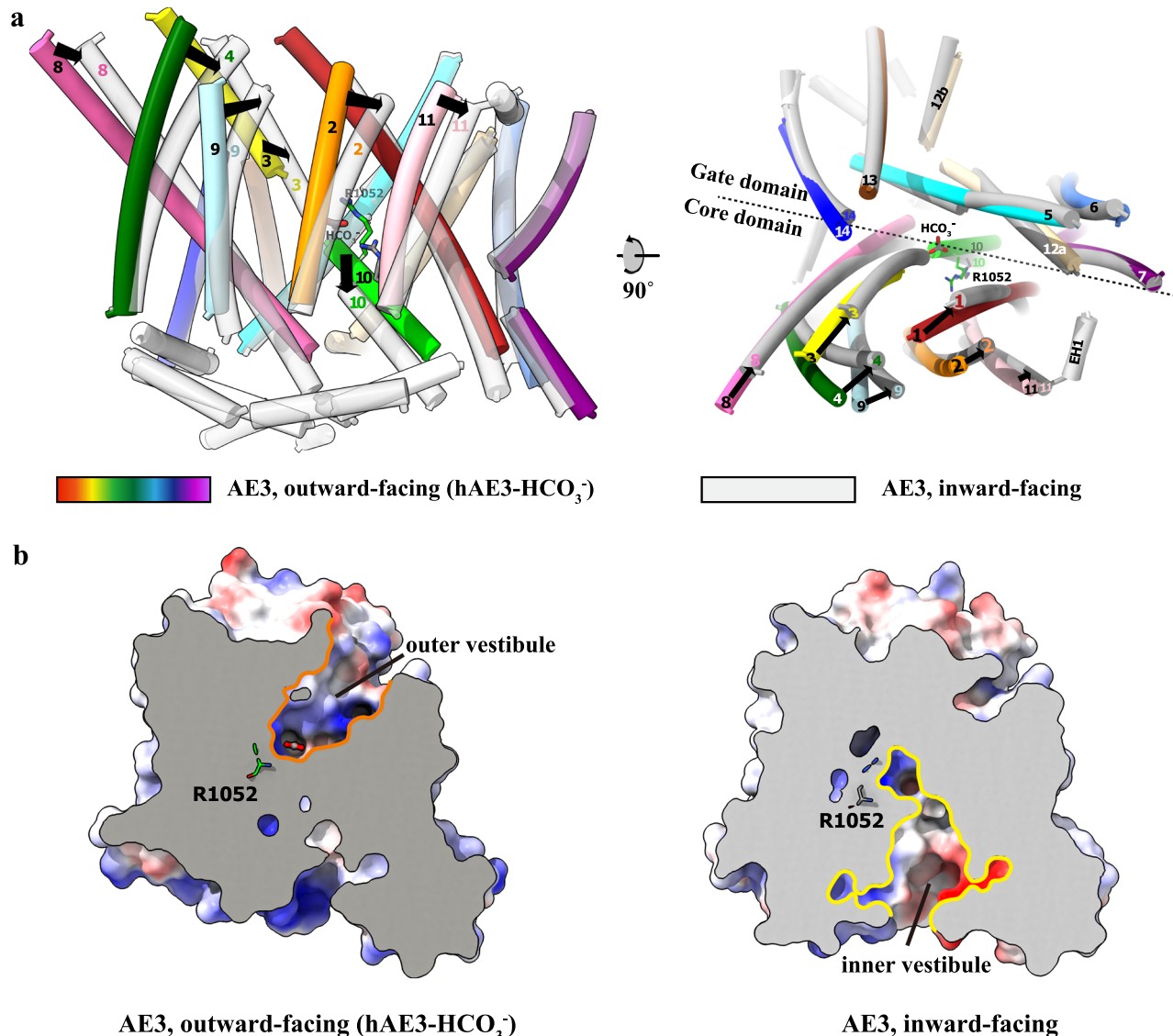

**a**

AE3, outward-facing (hAE3-HCO$_3$$^-$)          AE3, inward-facing

**b**

AE3, outward-facing (hAE3-HCO$_3$$^-$)          AE3, inward-facing

**Fig. 6 | Conformational Dynamics of hAE3 During Substrate Transport. a** The comparative analysis of the hAE3 outward- and inward-facing conformations. The cryo-EM structure of hAE3$^{HCO_3^-}$ was aligned with the homology -based model depicting its inward-facing state. The remarkable movements of the helices were indicated with arrows. **b** The outer and inner vestibules were shown in the AE3 structures in the outward- and inward-facing conformations, respectively. The cryo-EM structure of hAE3$^{HCO_3^-}$ and the homology-based model of AE3 in the inward-facing conformation were shown as the solvent-accessible electrostatic surface–potential maps, cut away to reveal the vestibules. The key substrate-binding residue, R1052, is depicted in a stick model, color-coded by atomic element, underscoring its role in substrate coordination.

role in lipid regulation of AE3 function. Thus, the R925H mutation could disrupt lipid-AE3 interactions, thereby affecting the modulation of AE3 and contributing to disease. Other critical mutation sites include E825 and S1012 within the inner vestibule, which may influence substrate entry and exit, and R343, R573, and R594 in the NTD[12,13,48], which could potentially disrupt global NTD folding when mutated.

Collectively, the structures of AE3 and AE3$^{NTD}$2$^{TMD}$ chimera, in combination with the functional assay and structural computation, revealed the overall structural folding and substrate-binding pocket for human AE3. The subtle structural distinctions between AE3 and AE2 enhance our understanding of their differential molecular functions and roles in pH homeostasis. Future dedicated efforts for solving the AE3 in alternative conformations and full-length are clearly warranted since it would enrich our understanding of anion exchange mechanisms and metabolic regulation, offering more profound insights into the nuanced interplay governing cellular pH balance.

## Methods

### Expression constructs

The cDNA encoding human Cardiac Anion Exchanger 3 (hAE3; UniProt ID: P48751-2) was a kind gift from Professor Han Jiahuai's laboratory at Xiamen University. This cDNA was subcloned into a modified pCAGGS mammalian expression vector (Addgene), which includes an N-terminal Flag tag, an 8 × His tag, and a Tobacco Etch Virus (TEV) protease cleavage site for enhanced expression and purification. The chimeric construct, hAE3$^{NTD}$2$^{TMD}$, was engineered to contain the N-terminal domain (NTD) of hAE3 (residues M297 to E671 as the numbering of UniProt ID P48751-1) fused to the C-terminal domain (TMD) of hAE2 (residues D672 to V1241 as the numbering of UniProt P04920-1 https://www.uniprot.org/uniprotkb/P48751/entry#P04920-1). The construction of the chimeric cDNA was achieved using overlap extension PCR and was subsequently inserted into the pCAGGS vector. Site-directed mutagenesis, facilitated by PCR, was employed to

introduce specific point mutations for the subsequent functional analysis. Prior to functional and structural experiments, all constructs underwent confirmation via DNA sequencing (Biosune-Biotech).

## Protein expression and purification

The recombinant protein was expressed by the suspension HEK293F cells (Thermo Fisher Scientific, A14527) cultured at 37 °C with 5% $CO_2$. The cell density was maintained at $1.0-4.0 \times 10^6$ cells per mL through cell passaging. Transient transfection was performed using PEI when cell density reached 2.5 and $3.0 \times 10^6$ cells/mL in a one-liter culture. Specifically, 1 mg of plasmid DNA and 3 mg PEI MAX (Polysciences) were each diluted in 50 mL of serum-free medium and incubated for 5 min before mixing. The mixture was allowed to stand for 25 min, then added to the cells and cultured for 72 h. Cell pellets were collected by centrifugation at 1800 g for 15 min and stored at −80 °C.

For protein purification, all procedures were performed at 4 °C or on ice. Frozen cell pellets were thawed, resuspended in hypotonic buffer (10 mM NaCl, 10 mM HEPES, pH 7.4, 1 mM PMSF), and lysed using a Dounce homogenizer. The lysate was centrifuged at 45,000 g for 30 min to pellet cell debris. The pellet was resuspended in hypertonic buffer (1 M NaCl, 25 mM HEPES, pH 7.4, 1 mM PMSF, 5 mM $MgCl_2$, 0.1 mg/mL DNase I), followed by another centrifugation step. The final membrane pellet was resuspended in Lysis Buffer (150 mM NaCl, 20 mM HEPES, pH 7.4, 1 mM PMSF, 5 mM $MgCl_2$, 0.1 mg/mL DNase I, 1× protease inhibitor cocktail) with 10× detergent buffer (1% LMNG, 0.1% cholesteryl hemisuccinate, 0.01% GDN) added, and stirred gently for 2 h. The solubilized membrane was centrifuged at 45,000 g for 45 min, and the supernatant was purified using anti-Flag affinity chromatography, followed by elution with 3×Flag peptide in the presence of 0.01% GDN.

The purified protein was subjected to size-exclusion chromatography on a Superose 6 Increase 10/300 GL column (GE Healthcare) in a buffer containing 150 mM NaCl, 20 mM HEPES, pH 7.4, and 0.01% GDN. Fractions corresponding to the homodimer were concentrated to -8 mg/mL for cryo-EM analysis. For AE3 bound to $NaHCO_3$ and DIDS, 50 mM $NaHCO_3$ and 200 μM DIDS were supplemented to the protein solution and incubated for 1 h at 4 °C before cryo-grid preparation.

## Cryo-EM sample preparation and data collection

The preparation of cryo-EM samples was conducted using a Thermo Fisher Vitrobot Mark IV (FEI) set at 8 °C and 100% humidity. Approximately 3 μL aliquots of the samples were applied to glow-discharged holey carbon grids (Quantifoil R1.2/1.3 Au, 300 mesh). After a 10-second incubation, the grids were blotted for 1 s using filter paper (Whatman) and immediately plunged into liquid ethane cooled by liquid nitrogen for rapid freezing. High-resolution cryo-sample images were collected using a 300 kV cryo-EM (FEI, Titan Krios) equipped with a K3 Summit direct electron detector (Gatan), a GIF Quantum energy filter (Gatan), and a Cs corrector (Thermo Fisher), operating in zero-energy-loss mode. Movie stacks were automatically collected using EPU software at a nominal magnification of 81,000 × (corresponding to a physical pixel size of 1.1 Å), with a defocus of approximately −1.8 μm. The dose rate was set to -14.9 electrons/$Å^2$ per second, and the total exposure time was 3.4 s, resulting in a total dose of 50 electrons/$Å^2$, fractionated into 32 frames.

## Cryo-EM data processing

Four datasets were collected: 4395 movie stacks for the hAE3 apo dataset, 3480 for the $hAE3^{HCO_3^-}$ dataset, 4000 for the $hAE3^{HCO_3^-/DIDS}$ dataset, 3783 for the $hAE3^{NTD2TMD}$ dataset. All cryo-EM data image processing steps were performed using cryoSPARC and RELION-3[49]. Motion correction was carried out on the dose-fractioned image stacks using MotionCor2 with 6 × 5 patches[49], and the contrast transfer function (CTF) parameters of each image were estimated using Gctf[50].

Initially, particles were automatically picked using Gautomatch-v0.56 (developed by Kai Zhang) without a template. Subsequently, particles were re-picked by Gautomatch-v0.56 when high-quality 2D or 3D references were generated. The particles were firstly extracted with 2× binning (2.2 Å/pixel) in RELION-3 and then subjected to cryoSPARC. Poor-quality particles were removed through one round of 2D classifications. Part of the selected particles was used for Ab-Initio Reconstruction to generate references for subsequent 3D classification. After multiple iterations of Heterogeneous Refinement, particles corresponding to the best class were re-extracted without binning (1.1 Å/pixel) and further processed with 3D auto-refinement, solvent-masked postprocessing, and Bayesian polishing in RELION-3. The polished particles were then re-subjected to cryoSPARC, followed by multiple iterations of Heterogeneous Refinement and Non-uniform Refinement with C2 symmetry until a high-resolution map was yielded. All resolutions were estimated using the gold-standard Fourier shell correlation 0.143 criteria with high-resolution noise substitution. Local resolution estimation was performed by Local Resolution Estimation. The detailed data information and processing procedures are illustrated in Table S1, Figures S2–S4, and S7.

## Model building and refinement

The cryo-EM structure models of $hAE2^{NaHCO_3}$ (PDB 8GVC) and $hAE2^{acidic-KNO_3}$ (PDB 8GVH) were employed as templates for the initial model building of the hAE3 apo form and the $hAE3^{NTD2TMD}$ chimera, respectively. These initial models were subjected to iterative rounds of manual adjustments using Real-space Refinement module of the Phenix package and COOT[51,52]. For the structures of $hAE3^{HCO_3^-}$ and $hAE3^{HCO_3^-/DIDS}$, the hAE3 apo model served as the initial model. The DIDS and $HCO_3^-$ molecules were fitted and refined using the LigandFit and Real-space Refinement modules of the Phenix package.

## RNA extraction and RT-qPCR assay

To isolate total RNA from HEK293F cells, 2 mL of the cell suspension, corresponding to approximately $4 \times 10^6$ cells, was centrifuged at 2000 g for 5 min. The resultant cell pellets were processed using the FastPure Cell/Tissue Total RNA Isolation Kit V2 (Vazyme) following the manufacturer's instructions to extract total RNA. For cDNA synthesis, about 1 ng of the extracted total RNA served as a template in a reverse transcription reaction using the Hifair III 1st Strand cDNA Synthesis SuperMix for qPCR Kit (Vazyme). The primers for Reverse transcription qPCR (RT-qPCR) were designed by the PrimerBank (https://pga.mgh.harvard.edu/primerbank/). RT-qPCR experiment was performed with Hieff qPCR SYBR Green Master Mix (Yeasen) on the LightCycler 480 Real-Time PCR System (Roche). The $2^{-\Delta\Delta CT}$ method was utilized to calculate the relative mRNA expression levels of *SLC4A1*, *SLC4A2*, and *SLC4A3* in HEK293 cells.

## CRISPR-Cas9 Knockout and anion exchange activity assay

The *SLC4A2* knockout (KO) HEK293F cell line used for measuring anion exchange activity was generated using CRISPR-Cas9 technology. The sgRNA sequences targeting *SLC4A2* were designed using CRISPOR. The forward and reverse primers were annealed and introduced into the lentiCRISPRv2GFP vector (Addgene, plasmid #82416), enabling co-expression of PuroR, Cas9, and the sgRNA sequence. Adherent HEK293F cells cultured in Dulbecco's modified Eagle's medium (DMEM, Sigma-Aldrich) supplemented with 10% fetal bovine serum (Wisent Corporation, catalog # 080–150) at 37 °C with 5% $CO_2$ were transfected with the plasmid using Lipofectamine 2000. After 48 h, the untransfected and transfected HEK293F cells were transferred to a selection medium containing puromycin. After 4–6 days, when the untransfected HEK293F cells were all dead, the transfected cells were collected and isolated as single cells into 96-well plates through Flow Cytometer (BD LSRFortessa). The disruption of the target gene in each monoclonal cell line was confirmed by Western blot. The validated *SLC4A2*-KO

HEK293F cell line was then adapted to suspension culture in a serum-free medium at 37 °C with 5% $CO_2$. The number of passages for this cell line was kept within 20, and cell line authentication was performed after the anion exchange activity assay (Supplementary Fig. 1b).

For the anion exchange activity assay, *SLC4A2*-KO HEK293F cells were transfected with the desired cDNAs and cultured for 48 h. For pH measurement, both untransfected and transfected cells were loaded with the intracellular fluorescent pH indicator BCECF-AM (Sigma-Aldrich) at 1 μg/mL in serum-free medium for 30 min at 37 °C. The cells were then resuspended in a $Cl^-$-free buffer (140 mM $Na^+$-Gluconate, 5 mM $K^+$-Gluconate, 1 mM $MgSO_4$, 1 mM $Ca^{2+}$-Gluconate, 5 mM Glucose, 25 mM $NaHCO_3$, 2.5 mM $NaH_2PO_4$, 10 mM HEPES, pH 7.4) and divided into two components. One component underwent fluorescence measurements at 488 nm excitation, and emission ratios at 530/661 nm were recorded using a Flow Cytometer after incubation for 10 min at room temperature, serving as the zero time point in the assay. The other component's buffer was then replaced with $Cl^-$-containing buffer (140 mM NaCl, 5 mM KCl, 1 mM MgSO4, 1 mM $Ca^{2+}$-Gluconate, 5 mM Glucose, 25 mM $NaHCO_3$, 2.5 mM NaH2PO4, 10 mM HEPES, pH 7.4) after 10 min of incubation and was applied to the Flow Cytometer. Readings were recorded every 40 s until 160 s. The intracellular pH was calculated using the standard curve generated by the Intracellular pH Calibration Buffer Kit (Thermo), where '*y*' represents the intracellular pH (pHi) and '*x*' represents the ratio of Em535/Em661, following the equation $y = 1.4794x + 4.0932$. The pHi-time curves and differences in pHi (delta pHi) were plotted and analyzed in GraphPad Prism, Statistical significance was assessed using One-way ANOVA with Tukey's multiple comparisons test.

## Western blot

For Western blot analysis, HEK293F cells, untreated *SLC4A2*-KO HEK293F cells, and cells from functional assays were harvested by centrifugation at 3200 g for 10 min. Subsequently, they were resuspended in Lysis Buffer (150 mM NaCl, 20 mM HEPES, pH 7.4, 1 mM PMSF, 5 mM $MgCl_2$, 0.1 mg/mL DNase I, 1 × protease inhibitor cocktail) and sonicated for 1 minute on ice. The whole-cell lysates were then solubilized by incubating with 1% (w/v) n-Dodecyl-β-D-Maltoside (DDM) (Anatrace) at 4 °C for 2 h. The solubilized lysates were centrifuged at 16,000 g for 10 min at 4 °C, and the supernatants were subjected to SDS-PAGE. Proteins separated on the gels were then transferred onto PVDF membranes (Merck/Millipore) using a semi-dry transfer system (Bio-Rad). Membranes were blocked with 5% (w/v) skim milk (BD Difco) for 2 h at room temperature, followed by incubation with primary antibodies diluted in 2.5% (w/v) skim milk for 2 h. The primary antibodies used were Mouse anti-DDDDK-Tag mAb (ABclonal, AE005), SLC4A2 Rabbit pAb (ABclonal, A7729), and GAPDH Rabbit mAb (ABclonal, A19056). This was followed by incubation with HRP-linked secondary antibodies in 2.5% (w/v) skim milk for 1 h; the secondary antibodies were Anti-mouse IgG, HRP-linked Antibody (Cell Signaling, #7076), and HRP-linked Antibody (TransGen Biotech, Catalog number: HS101-01). Detection was performed using ECL Prime Western Blotting Detection Reagent (Cytiva) and visualized with an Amersham Imager 600 (GE Healthcare).

## Reporting summary

Further information on research design is available in the Nature Portfolio Reporting Summary linked to this article.

## Data availability

The coordinates have been deposited in the PDB with the accession codes 8Y85, 8Y86, 8Y8K, and 8ZLE. The cryo-EM maps have been deposited in the Electron Microscopy Data Bank (EMDB) with accession codes EMDB-39034 [https://www.ebi.ac.uk/pdbe/entry/emdb/EMD-39034], EMDB-39035 [https://www.ebi.ac.uk/pdbe/entry/emdb/EMD-39035], 39050, and 60225. Source data are provided with this paper.

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

## Acknowledgements

The authors thank Drs. Ming Zhou, Ming Lei, and Lijun Wang (for scientific discussion); Dr. Rijing Liao (for assistance with the data analysis). This work was supported by the National Natural Science Foundation of China (82072468 and 82272519, Y. C.; 82371067, L. L.; 92068102 and 82372430, A. Q.), Shanghai Municipal Committee of Science and Technology (20S11902000, Y.C), Shanghai Key Laboratory of Orthopedic Implants KFKT202207 (A. Q.). This work was also supported by the Shanghai Frontiers Science Center of Degeneration and Regeneration in Skeletal System (A. Q.) and the Innovative Research Team of High-level Local Universities (SHSMU-ZLCX20211700, Y. C.) from the Shanghai Municipal Education Commission. We thank the staff members of the Electron Microimaging Center, Bioimaging Facility, and Proteomics Platform at the Shanghai Institute of Precision Medicine for providing technical support and assistance in data collection.

## Author contributions

Y.C. and L.L. conceived the study. Y.C., L.L., L.J., Q.Z., A.Q., and S.L. designed the experiments. L.J., Q.Z., M.Ch., and Y.X. performed the functional assay. Q.Z., L.J., M.Ca., Q.W., Y.S., and Y.C. performed structural biology experiments. Y.C. and D.Y. built and refined structural models. Y.C., L.J., Q.Z., D.Y., L.L., and A.Q. wrote the manuscript.

## Competing interests

The authors declare no competing interests.
