## [Peer Review File · Nature Communications]

The structural insight into the functional modulation of human anion exchanger 3REVIEWER COMMENTS

Reviewer #1 (Remarks to the Author):

The authors report multiple cryoEM structures of the transmembrane domain homodimer of human cardiac isoform of AE3/SLC4A3, the least studied of the Na⁺-independent members of the SLC4 gene family. The structures arrive after publication of structures for AE1 and AE2. Because they found the AE3 N-terminal cytoplasmic domain unstructured in the holoprotein, the authors also determined the structure of a chimeric AE3NTD/AE2TMD homodimer.

Abstract:

The second half of the abstract is vague, and would be better written to address specific points.

Introductory section:

Many errors or mis/overstatements (major as well as minor) include the following:

Line 6: change "carbonate" to "bicarbonate" or to "CO₂ and bicarbonate"

Line 8: change "sodium efflux" to "sodium influx". Na-dependent chloride/bicarbonate exchange usually mediates bicarbonate influx.

Lines 14-15 cite elevated cellular pH in zebrafish heart (ref 13), but no pH change has been documented to my knowledge in brain or retina. Authors should cite literature in support of their statement. It seems absent from ref 13.

Line 24: SLC4A4-10 are not all sodium-dependent cotransporters, but include sodium dependent anion exchangers.

Line 38: AE3 anion exchange is not particularly "distinct", but is distinctly regulated.

Line 40: Specify that the lower activity of AE3 in the cited reports is in recombinant expression systems.

Methods:

Expression constructs

p.20 line 39: Provide 1 hour incubation temperature with 200 microM DIDS.

Cell-based anion exchange activity assay p. 22 line 12.

Provide method of construction of SLC4A2 CRISPR/Cas9 knockout in HEK 293F cells and selection medium and growth medium conditions of knockout line(s). Cite Supp. Fig 1B as validation data for absence of AE2 protein in knockout line. Note passage number of analyzed cells and document phenotypic stability of AE2 knockout line.

Sample preparation for cryoEM

P 20 lines 36-39: Clarify if 50 mM NaHCO₃ and 200 microM DIDS were added to 150 mM NaCl +20 mM HEPES. Whether or not the case, what was the pH of the solution containing 50 mM NaHCO₃. At what temperature were the components incubated for 1 hr?

pH assay

p. 22 lines 18-23. Incubation buffers were CO₂-free. What was pH of samples after 10 min room temperature in Cl⁻ free buffer (zero time in assay)? And what was pH at end-assay just prior to FACS measurement of pH?

p. 22 lines 24-25: pH measurements are taken at 40 sec intervals (Fig 1a). Could sequential samples be applied to the FACS every 40 sec with time for interim washout and resetting?

Western blot

p. 22 line 33: for sample prep for western blot, were there any fractionation steps between sonication and DDM solubilization? Authors should specify if nuclear DNA was removed prior to DDM solubilization.

Results:

p. 3 line 14: "the primary pH controller" is incorrect. Change to "the primary anion exchanger" or "primary acid loader".

Fig 1b legend – Specify DIDS concentration used for AE2 and for AE3 functional assay.

Supp Fig 1c,d panels and legend: specify fraction #s pooled for running on adjacent protein gels. The elution profiles are not very homogeneous – do they represent merely differential states of aggregation of intact protein? The protein gel in panel Supp Fig 1d shows moderate degradation. Was that a typical CryoEM prep?

p. 3 line 38: provide more detail about the salt and ligand conditions and vectorial state (outward-facing vs inward-facing) for reference cryo-EM structure of AE2 which served as starting model Fig. 4 and p. 5 lines 8-17: Detection of putatively covalently bound DIDS together with bound HCO₃⁻. Simultaneous binding of DIDS together with substrate anion HCO₃⁻ is to my knowledge unprecedented. Classically, the stilbene sulfonate moiety has been thought to destabilize the bound bicarbonate, and vice-versa. This structure needs to be discussed in comparison with the APO structure in absence of substrate anion, and in comparison with and re-evaluation of published DIDS-liganded AE1 structures not reported to contain bound substrate anion. What are the consequences to the currently accepted alternating access transport model. How can simultaneous binding of HCO₃ and DIDS be fit to the elevator model of transport? As the structure backbones are very similar, it ought to be possible to dock DIDS into the AE3-HCO₃ structure so as to "yield" the equivalent of the AE3-HCO₃-DIDS structure. Is this the case?

p. 7 lines 5-6 and Discussion. What about the AE3 sequence and/or structure favors the predominance of the TMD outward-facing conformation, in contrast to AE2 and AE1?

Discussion:

Authors do not comment about lipid content of particles, presence or absence of either cholesterol, PIP₂ or other lipids, which have been shown regulatory for AE2 and/or AE1. This discussion is needed. Authors do not comment on the AE3 TMD residues that correspond to the PIP₂-binding residues that in AE2 have been implicated in anion exchange regulation by changing pH. This discussion is needed, and mutation to the AE2 residues would be of interest to demonstrate gain-of-regulatory function by pH change.

P. 7 lines 19-21 – "capacity to lower acidity in response to significant alkalosis challenges in acid-secreting tissue" – this phrase is confusing.

Lines 22-23 – what is meant by "In these environments [of excitable tissues], pH variations are more subtly tied to metabolic processes?" This phrase seems confused and unjustified – in fact it's easier to image more extreme changes rather than gradual and moderate changes in excitable tissues.

Line 30: change "carbonate" to "bicarbonate"

Lines 30-32: discussion of AE2 inner vestibular occupancy by C-terminal loop requires citation.

Minor points;

P 2 Line 7: change "were evolved" to "evolved"

P2 Line 23: "transportation" should be "transport."

P 4 line 1 "adapts into" change to "adopts a homodimeric state"

P 4 line 2: change "soluble" to "cytoplasmic"

p. 22 line 16: change "stained" to "loaded"

Reviewer #2 (Remarks to the Author):

In this manuscript by Jian L et al. the authors reveal the high resolution cryo-EM structure of human Anion Exchanger-3 (SLC4A3) bound to carbonate and the small molecule inhibitor DIDS. Revealing the structure of hAE3 is timely, as recent publications have also highlighted structures of hAE1 and hAE2. Thus, this manuscript provides one of the final missing pieces in the structural puzzle surrounding this family of anion exchangers. Moreover, the structures of hAE3 presented here provide structural context from which to interpret the different pH and DIDS sensitivities of transporters in this family.

In contrast to previous investigations with hAE1 and 2, the authors found that hAE3 predominately adopts the outward facing conformation, and attempts to obtain an inward facing structure of hAE3 were unsuccessful. Furthermore, the NTD soluble domains of hAE3 display significant flexibility in the outward facing structures of hAE3. To overcome these limitations the authors utilized previous inward facing structures to construct a (likely accurate) homology model of hAE3 in an inward facing conformation, and also constructed a hAE3(NTD)2(TMD) chimera to facilitate structural elucidation of the hAE3 flexible NTD soluble domain. The approaches utilized in this manuscript are novel, and provide critical insight into this important family of human Anion Exchangers.

Overall, the manuscript is well written, the figures are well designed and clear to aid the reader in interpretation, and all data both structural and functional appears to be of high quality. I have only limited points for the authors to consider before publication...

1. One part of the manuscript that I struggled with begins on line 167 (describing the anion binding pocket). I believe there are several errors in the numbering of residues in this paragraph. I believe the authors want to refer to residue K1173 throughout the paragraph. But this residue is misnumbered as K1133 (line 167) and K1153 (line 184). Please confirm that all numbering is consistent.
2. On line 234 and 238, I believe the authors should point to supplemental figure 8B instead of 8A.
3. Throughout the manuscript the authors refer to the cryo-EM map as "electron density". Cryo-EM maps are not electron density, they are a representative map of the coulomb potential of the protein (seeMarques MA, Purdy MD, Yeager M. CryoEM maps are full of potential. Curr Opin Struct Biol. 2019. PMID: 31400843). I would recommend to avoid referring to cryo-EM maps as electron density.
4. In supplemental figure 6B it may be nice to show the map for the bound HCO₃⁻ if possible.
5. Figure 1E, the symmetry operator symbol showing the two-fold axis appears to be off centered.
6. In Supplemental table 1 under the refinement section, the model resolution should be reported at a cutoff of 0.5 (not 0.143).

Reviewer #3 (Remarks to the Author):

The manuscript by Jian and Cao et al reports structures of human anion exchanger 3 in the apo state, in the presence of a substrate, and in the presence of both the substrate and an inhibitor DIDS. Since the NTD of AE3 was not resolved the three structures, the authors determined the structure of a chimeric transporter with the NTD of AE3 and TMD of AE2. The structure of the chimera shows that the structure of AE3 NTD is similar to these of AE1 and AE3, and that the interactions between AE3 NTD to AE2 TMD are comparable to these of AE2 NTD and TMD. However, the strength of NTD/TMD interactions in AE3 could be weaker due to differences in amino acid sequences. The authors concluded that AE3 tends to stay in the outward facing conformation and that this preference is the cause for its higher sensitivity to DIDS and slower rate of transport and thus modest activity in pH

regulation.

I feel that the structures are well done and these structures are part of our knowledge base for understanding of AE family of transporters. I am not convinced that the lack of AE3 in the inward facing conformation on EM grids is sufficient evidence for its higher sensitivity to the inhibitor nor its slower rate of transport than the other AEs. If the authors are enthusiastic about these conclusions, then more precise experiments are needed, however, I should clarify that the comment is not a demand for more experiments.

The authors may want to change "electron density" to "density". The former is reserved for X-ray diffraction methods.

Reviewer #1 (Remarks to the Author):

The authors report multiple cryoEM structures of the transmembrane domain homodimer of human cardiac isoform of AE3/SLC4A3, the least studied of the Na⁺-independent members of the SLC4 gene family. The structures arrive after publication of structures for AE1 and AE2. Because they found the AE3 N-terminal cytoplasmic domain unstructured in the holoprotein, the authors also determined the structure of a chimeric AE3NTD/AE2TMD homodimer.

Abstract:

The second half of the abstract is vague and would be better written to address specific points.

Our response: We thank the reviewer for the feedback regarding the clarity of our abstract. We have revised the abstract to include specific details about the structural discoveries presented in our manuscript. These revisions clarify the significant differences between AE3 and AE2, particularly in their conformational preferences and interactions, which directly impact their functional roles in cellular environments. We believe these changes will greatly enhance the reader's understanding of our study's contributions to the field.

Introductory section: Many errors or mis/overstatements (major as well as minor) include the following:

Line 6: change “carbonate” to “bicarbonate” or to “CO₂ and bicarbonate”

Our response: We appreciate the suggestion regarding the terminology in the introductory section. As recommended, we have revised the text thoroughly to ensure scientific accuracy and clarity, including replacing "carbonate" with "CO₂ and bicarbonate".

Line 8: change “sodium efflux” to “sodium influx”. Na-dependent chloride/bicarbonate exchange usually mediates bicarbonate influx.

Our response: Thank you for the suggestion, we have changed “sodium efflux” to “sodium influx” in accordance with the main function of Na-dependent chloride/bicarbonate exchangers.

Lines 14-15 cite elevated cellular pH in zebrafish heart (ref 13), but no pH change has been documented to my knowledge in brain or retina. Authors should cite literature in support of their statement. It seems absent from ref 13.

Our response: We appreciate the reviewer's discussion on pH changes in the heart, brain, and retina. The knockdown of *slc4a3* in zebrafish heart, leading to elevated cellular pH, was also reported by M. K. Christiansen's research ¹, which we have cited in our manuscript as recommended. Upon careful review of related literature, we found that the changes in intracellular pH are ambiguous in the brain or retina when SLC4A3 is knocked down or mutated ². Moreover, only a slight tendency toward increased

steady-state pHi was observed in mice with AE3 knockout neurons³. Thus, we changed "lead to an elevated cellular pH" to "lead to disturbances in pH homeostasis" and provided more accurate descriptions about the diseases caused by the disruption of AE3 in the heart and brain.

Line 24: SLC4A4-10 are not all sodium-dependent cotransporters, but include sodium dependent anion exchangers.

Our response: We thank the reviewer for pointing out that SLC4A4-10 are not all sodium-dependent cotransporters. Some findings indicate that SLC4A9 operates as a sodium-dependent transporter, while it has also been found to function in a sodium-independent manner^{4,5}. Consequently, the definitive function of SLC4A9 still remains controversial, and assigning it as a sodium-dependent transporter lacks precision. Accordingly, we have revised the text to clarify the classification of the SLC4 family.

Line 38: AE3 anion exchange is not particularly "distinct", but is distinctly regulated.

Our response: We thank the reviewer for pointed that AE3 anion exchange is not particularly "distinct" but is "distinctly regulated." We have revised the text to be more accurate.

Line 40: Specify that the lower activity of AE3 in the cited reports is in recombinant expression systems.

Our response: We thank the reviewer for the suggestion to specify that the lower activity of AE3 in the cited reports is observed in recombinant expression systems. We have introduced this clarification as advised.

Methods:

Expression constructs

p.20 line 39: Provide 1 hour incubation temperature with 200 microM DIDS.

Our response: Thank you for the suggestion. The protein solution incubated with substrate or inhibitor was at 4 °C. We have added this information in the Method part.

Cell-based anion exchange activity assay p. 22 line 12.

Provide method of construction of SLC4A2 CRISPR/Cas9 knockout in HEK 293F cells and selection medium and growth medium conditions of knockout line(s). Cite Supp. Fig 1B as validation data for absence of AE2 protein in knockout line. Note passage number of analyzed cells and document phenotypic stability of AE2 knockout line.

Our response: Thank you for the suggestion that the method of constructing the SLC4A2 CRISPR/Cas9 knockout in HEK 293F cells should be introduced and the measures needed to maintain phenotypic stability. We have described the detailed process of generating the SLC4A2 CRISPR/Cas9 knockout HEK293F cell line, including sgRNA sequence design, plasmid construction, selection method, and growth conditions. To maintain phenotypic stability, the number of passages was kept within 20, and cell line authentication was performed. The validation data for the absence of AE2 protein in the

knockout line (Supplementary Figure 1b) was obtained after the anion exchange activity assay and is also cited in the corrected manuscript as recommended.

Sample preparation for cryoEM

P 20 lines 36-39: Clarify if 50 mM NaHCO₃ and 200 microM DIDS were added to 150 mM NaCl +20 mM HEPES. Whether or not the case, what was the pH of the solution containing 50 mM NaHCO₃. At what temperature were the components incubated for 1 hr?

Our response: We appreciate the reviewer's inquiry regarding whether 50 mM NaHCO₃ and 200 μM DIDS were added to the buffer (150 mM NaCl and 20 mM HEPES, pH 7.4). When the primary affinity-purified protein was subjected to size-exclusion chromatography, the FPLC buffer did not include NaHCO₃ and DIDS. Instead, NaHCO₃ and DIDS were supplemented and incubated for 1 hour at 4 °C before cryo-grid preparation. We measured the pH values when the solution was supplemented with 50 mM NaHCO₃, it was approximately 7.8 at 4 °C. In response to this, we have provided a more detailed description of this preparation process and the incubation temperature in the manuscript, as suggested.

pH assay

p. 22 lines 18-23. Incubation buffers were CO₂-free. What was pH of samples after 10 min room temperature in Cl⁻ free buffer (zero time in assay)? And what was pH at end-assay just prior to FACS measurement of pH?

Our response: We appreciate the reviewer's inquiry regarding the pH during the activity assay. The pH of the samples after incubation for 10 minutes at room temperature in Cl⁻-free buffer are indicated at the zero time point in the assay. The pH just prior to FACS measurement at the end of the assay is indicated at the 160-second point in the assay. To improve clarity and comprehension, we have provided a more detailed description of the pH assay process in the manuscript.

p. 22 lines 24-25: pH measurements are taken at 40 sec intervals (Fig 1a). Could sequential samples be applied to the FACS every 40 sec with time for interim washout and resetting?

Our response: Thank you for inquiring about the pH assay process. When the sample is changed to a Cl⁻-containing buffer and applied to the Flow Cytometer, cells with changing intracellular pH continuously enter the machine and are recorded every 40 seconds until 160 seconds. The process is continuous, so interim washout and resetting are unnecessary. We have included a more detailed description of the pH assay methods in the manuscript to improve clarity and comprehension of this process.

Western blot

p. 22 line 33: for sample prep for western blot, were there any fractionation steps between sonication and DDM solubilization? Authors should specify if nuclear DNA was removed prior to DDM solubilization.

Our response: We appreciate the reviewer's inquiry about whether any fractionation steps were conducted between sonication and DDM solubilization. When the cells, which were resuspended in lysis buffer, were sonicated, no fractionation steps were performed. The whole-cell lysates were subsequently solubilized by DDM. The nuclear DNA was disrupted by DNase application during DDM solubilization and was not removed by centrifugation prior to DDM solubilization. We have revised the Methods of Western blot in the manuscript to provide a more detailed description and further clarify the process.

Results:

p. 3 line 14: "the primary pH controller" is incorrect. Change to "the primary anion exchanger" or "primary acid loader".

Our response: Thank the reviewer for pointed that "the primary pH controller" is inappropriate on p. 3 line 14. We have changed change to "the primary anion exchanger" as recommended.

Fig 1b legend – Specify DIDS concentration used for AE2 and for AE3 functional assay.

Our response: Thank you for the suggestion. The DIDS concentration used for AE2 and AE3 functional assays was 40 μ M. We have added the concentration information as recommended.

Supp Fig 1c,d panels and legend: specify fraction #s pooled for running on adjacent protein gels. The elution profiles are not very homogeneous – do they represent merely differential states of aggregation of intact protein? The protein gel in panel Supp Fig 1d shows moderate degradation. Was that a typical CryoEM prep?

Our response: Thank you for the suggestion. We have updated Supplementary Figure 1c and 1d to specify the fraction used for SDS-PAGE. The elution profiles indeed show three peaks. The first peak, with the lowest elution volume, typically represents macromolecular aggregations. The third peak, with the highest elution volume, corresponds to a homogeneous dimer which was used for cryo-EM to obtain the high-quality maps presented in this manuscript. The middle peak predominantly contains higher oligomeric states of the protein.

The peak in the middle mainly consists of higher oligomeric states of the protein. Cryo-EM analysis indicated significant heterogeneity in the higher oligomeric peak of AE3, so we didn't collect the data. The peak of the AE3^{NTD}2^{TMD} chimera exhibits homogeneity, 2D and 3D classification of this sample suggested a tetrameric arrangement, where two dimeric units interact via their NTDs (Fig. R1 below). Given the soluble domain-mediated interactions in this tetramer and the geometric constraints imposed by the TMDs embedded in the cell membrane, it is likely that this tetramer represents an artifact of sample preparation. Unfortunately, despite attempts at local refinement, the flexibility of the interactional NTDs prevented us from obtaining high-quality EM signals sufficient for detailed structural resolution.

Upon reviewing our experimental records, we found that the protein gel in Supplementary Figure 1d, which shows moderate degradation, wasn't a typical sample prepared for Cryo-EM. This degradation may have been caused by exposure to room temperature for too long before SDS-PAGE. Therefore, we

have replaced it with another set of FPLC results of AE3^{NTD2}TMD in Supplementary Figure 1d to represent a more accurate state for Cryo-EM.

Figure R1. The cryo-EM analysis of the higher oligomeric peak of AE3^{NTD2}TMD. **a**, A representative 2D class average. **b**, A representative 3D map with different views is displayed, shown with the contour level of 0.2. **c**, A representative 3D map shown with the contour level of 0.5.

p. 3 line 38: provide more detail about the salt and ligand conditions and vectorial state (outward-facing vs inward-facing) for reference cryo-EM structure of AE2 which served as starting model

Our response: Thank you for suggesting that we should provide more detailed information about the starting model. The initial model represents an outward-facing conformation of AE2 bound with HCO₃⁻. We have included this information in the text as recommended.

Fig. 4 and p. 5 lines 8-17: Detection of putatively covalently bound DIDS together with bound HCO₃⁻. Simultaneous binding of DIDS together with substrate anion HCO₃⁻ is to my knowledge unprecedented. Classically, the stilbene sulfonate moiety has been thought to destabilize the bound bicarbonate, and vice-versa. This structure needs to be discussed in comparison with the APO structure in absence of substrate anion, and in comparison with and re-evaluation of published DIDS-liganded AE1 structures not reported to contain bound substrate anion. What are the consequences to the currently accepted alternating access transport model. How can simultaneous binding of HCO₃⁻ and DIDS be fit to the elevator model of transport? As the structure backbones are very similar, it ought to be possible to dock DIDS into the AE3-HCO₃ structure so as to “yield” the equivalent of the AE3-HCO₃-DIDS structure. Is this the case?

Our response: We thank the reviewer for the question on the AE3 structure co-occupied by DIDS and HCO_3^- . In determining structures for AE3 structure bound with DIDS, our initial attempts at resolving the AE3 structure with DIDS did not reveal any density for DIDS at its binding site. It was only after extensive optimization of our sample preparation conditions that we observed EM signals for DIDS in structures supplemented with both DIDS and HCO_3^- . The resulting cryo-EM map, resolved to 2.89 Å, displayed non-protein densities at both the DIDS binding site and the substrate-binding site, supporting the modeling of AE3 in a conformation simultaneously bound with DIDS/ HCO_3^- . In previously documented AE family structures containing DIDS or H_2DIDS , such as the crystal structure of AE1-DIDS (PDB ID 4YZF), the cryo-EM structures of AE1-DIDS (PDB ID 8T6V), AE1- H_2DIDS (PDB ID 7TY6), and AE2-DIDS (PDB ID 8GV8)⁶⁻⁸, an empty substrate-binding site is typically observed. The inhibitory effect of DIDS/ H_2DIDS on substrate association in AE1 or 2 might be attributed to charge repulsion, as indicated by the proximity—3.8 Å and 4.7 Å—between the sulfonic acid groups of DIDS and the bicarbonate-binding sites in AE1 and AE2, respectively (Figure R2a). In contrast, in the $\text{hAE3}^{\text{HCO}_3^-/\text{DIDS}}$ structure, the distance between the sulfonic acid groups of DIDS and the bicarbonate ion exceeds 5.4 Å, allowing for their simultaneous presence within the outer vestibule. This structural arrangement underscores the unique interaction dynamics within AE3 compared to its counterparts, contributing to its distinct functional and inhibitory response profiles.

In a structural superposition between $\text{hAE3}^{\text{HCO}_3^-}$ and $\text{hAE3}^{\text{HCO}_3^-/\text{DIDS}}$, as shown in Figure R2 below, we didn't observe a significant variation in the overall folding and local conformation surrounding the DIDS and substrate binding sites, contradicting the classical view of the destabilizing effects of DIDS on bicarbonate binding in AE1. This suggests that DIDS can be accommodated within the $\text{hAE3}^{\text{HCO}_3^-}$ structure so as to “yield” the equivalent of the $\text{hAE3}^{\text{HCO}_3^-/\text{DIDS}}$ structure, as evidenced by the detailed comparison of the binding pockets (Fig. R2b, right panel).

We have expanded the discussion in the revised manuscript to address these findings and their implications for the inhibitory mechanism of AE3. We thank the reviewer for significantly enriching the discussion and highlighting the unique mechanistic implications of our findings on $\text{hAE3}^{\text{HCO}_3^-/\text{DIDS}}$ structure, and this really enhances the depth and quality of our manuscript.

Figure R2. The structural comparison among DIDS bound in AEs. **a.** The DIDS bound in the outer vestibules of AE3 (left), AE1 (middle), and AE2 (right). The AE1-3 structures were shown as cartoon model, with the DIDS and HCO₃⁻ shown as stick models. The HCO₃⁻ binding sites in AE1 and 2 were indicated with transparent stick model. **b.** The structural superposition between hAE3^{HCO₃⁻} and hAE3^{HCO₃⁻/DIDS}. Left: the structural superposition between hAE3^{HCO₃⁻} and hAE3^{HCO₃⁻/DIDS}. Both structures were shown as cartoon model, with hAE3^{HCO₃⁻} colored by helices in a rainbow gradient and hAE3^{HCO₃⁻/DIDS} colored in gray. The HCO₃⁻ and DIDS were shown as stick model. Right: Enlarged view of the binding pocket of DIDS in the structural superposition between hAE3^{HCO₃⁻} and hAE3^{HCO₃⁻/DIDS}. The key interacting residues were shown as stick model and colored as their respective backbone.

p. 7 lines 5-6 and Discussion. What about the AE3 sequence and/or structure favors the predominance of the TMD outward-facing conformation, in contrast to AE2 and AE1?

Our response: We appreciate the reviewer's question regarding the factors that favor the predominance of the outward-facing conformation in the transmembrane domain (TMD) of AE3, in contrast to AE2 and AE1. We would like to clarify that both AE1 and AE3 predominantly adopt an outward-facing conformation, whereas AE2 typically rests in an inward-facing conformation.

As detailed in the manuscript, key electrostatic interactions and hydrogen bonds integral to AE2's interlock, specifically E508-K1073, Q618-D1079, and E621-G1078, are absent in AE3 due to amino acid variations within the loop TMH10/11 and loop NH7/8. This lack of specific interactions in AE3 contributes to its preference for the outward-facing state. Furthermore, sequence alignment with AE1 reveals that the interactions Q618-D1079 and E621-G1078 are also absent in AE1. This observation is consistent with the current structural data for human AE1, where no inward-facing state has been resolved. This comparative analysis underscores the structural basis for the differing conformational predominance among AE family members.

We appreciate the reviewer's suggestion to elucidate these distinctions further. Consequently, we have expanded the Discussion section to address the conformational preferences of AE1, AE2, and AE3 and to clarify the structural basis underlying these differences.

Figure R3. A representative protein sequence alignment among AE2, AE3, and AE1 from *Homo sapiens*. The upper panel shows the sequence alignment of AE1-3 NTDs, and the lower panel shows the sequence alignment of AE1-3 loop^{TMD10/11}. The interacting residues in NTD-TMD interface were indicated by blue frames, with the blue lines connecting the residue pairs with electrostatic interactions and hydrogen bonds.

Discussion:

Authors do not comment about lipid content of particles, presence or absence of either cholesterol, PIP₂ or other lipids, which have been shown regulatory for AE2 and/or AE1. This discussion is needed.

Our response: Thank you for the suggestion that we should discuss the lipid content of particles. In our structure of the NTD^{AE3}-TMD^{AE2} chimera, we observed densities corresponding to PIP₂, cholesterol, and CHS in the TMD of AE2 (Fig. R4a), aligning with findings reported in previous models of AE1 and AE2⁸⁻¹⁰. This observation reinforces the role of lipids in regulating human AE2.

Furthermore, we conducted a comparative analysis of the PIP₂, cholesterol, and CHS binding sites across both AE2 and AE3 to deepen our understanding of lipid-TMD interactions. Notably, in AE3, while we detected dispersed densities at the PIP₂ binding site, cholesterol and CHS were absent. The densities at the PIP₂ site appeared fragmented and lacked sufficient structural detail to definitively model the inositol moiety of PIP₂ (Fig. R4b-4d). Given the ambiguous molecular identity, we chose not to model any lipid molecules at this site in AE3. Nevertheless, our structural analysis of the local structures corresponding to the PIP₂-binding site in AE2 revealed that the positively charged residues near the inositol phosphate group are largely conserved in AE3, including residues such as R925/932/933 and K1147 in AE2, and their counterparts in AE3 (Figures R4e and R4f). This conservation of a positively charged pocket suggests the potential presence of an acidic phospholipid at the cytoplasmic side of the dimeric interface, possibly playing a role similar to PIP₂ in AE2 modulation.

We have included a comprehensive discussion of these findings in the revised manuscript, specifically in the Discussion section. This includes an analysis of the implications of lipid interactions with AE3 and how they compare to those observed in AE2, further elucidated in Supplementary Figure 11 and supported by relevant literature citations.

Figure R4. The lipid bound in AE2 and AE3. a-d, The non-proteinous densities identified in the outer surfaces of the TMDs from AE2 and AE3 structures. The protein density maps were shown for hAE3^{NTD2TMD} (a), hAE3 apo (b), hAE3^{HCO₃⁻} (c), and hAE3^{HCO₃⁻/DIDS} (d), with the non-proteinous densities highlighted in orange for PIP2, green for cholesterol, pink for CHS, and transparent gray for other unidentified lipids. **e and f,** The enlarged view of the PIP2 binding sites in hAE3^{NTD2TMD} (e) and hAE3^{HCO₃⁻/DIDS} (f). The TMDs of AE2 and AE3 were shown as cartoon model. The PIP2 molecule and the contacting residues were shown as stick models. The non-proteinous densities in AE3 were shown as an orange surface model and set to transparent.

Authors do not comment on the AE3 TMD residues that correspond to the PIP2-binding residues that in AE2 have been implicated in anion exchange regulation by changing pH. This discussion is needed, and mutation to the AE2 residues would be of interest to demonstrate gain-of-regulatory function by pH change.

Our response: We thank the question and comments about the PIP2 binding residues in AE2 and AE3. As discussed in our response to a previous query, while the AE3 cryo-EM map density did not sufficiently support the precise modeling of PIP2 or other lipids at the position corresponding to the

PIP₂ binding pocket of AE2, there was non-proteinous density suggesting the presence of a lipid molecule. Our structural analysis revealed that key inositol phosphate-interacting residues in AE2, such as R925/932/933, K1147, and H1148, are highly conserved in AE3 (Fig. R4e and 4f). This conservation suggests that AE3 might similarly bind an acidic lipid, potentially playing a role analogous to PIP₂ in AE2 modulation. This implies that AE3 could be regulated by pH changes in a manner similar to AE2, due to these conserved residues. Future experiments, such as mutating these residues in AE3, are warranted to provide deeper insights into the regulatory mechanisms of AE3. We have included this discussion in the revised manuscript, highlighting the potential regulatory role of these conserved residues and the implications for anion exchange modulation.

The authors should discuss (and show in one of the existing figures) the topographical locations of the few human mutations in AE3 associated with either epilepsy or short QT syndrome."

Our response: We thank the reviewer and acknowledge significance of understanding the structural basis of these pathogenic mutations to further elucidate AE3-related disease mechanisms. We have thereby incorporated an analysis of several known epilepsy and short QT-associated mutations within the AE3 structure in the outward-facing conformation (Fig. R5 below), which is generated by merging the TMD of hAE3^{HCO₃⁻} model and the NTD of the hAE3^{NTD}2^{TMD} model. Additionally, we have included the PIP₂ potentially bound within this structure to enhance our discussion of lipid interactions and their possible regulatory roles. One notable mutation site is R925¹, located on the cytoplasmic side of the TMDs near the potential binding pocket for the hydrophilic head of phospholipid, such as the inositol phosphate moiety of PIP₂. Although our previous studies did not confirm the identity of the phospholipid in the AE3 TMD, the proximity of R925 to this region suggests its potential role in lipid regulation of AE3 function. Other critical mutagenesis sites include E825 and S1012 within the inner vestibule, which may influence substrate entry and exit, and R343, R573, and R594 in the NTD^{1,11,12}, potentially affecting global NTD folding when mutated.

We have detailed these pathogenic mutations in the discussion section of our manuscript and illustrated them in the supplementary Figure 12. Additionally, we have highlighted these disease-associated mutation sites in the topological diagram of the hAE3 monomer (Figure 1f), facilitating a clearer understanding of their implications for AE3 function and pathology.

Figure R5. The locations of the pathogenic mutation sites in AE3. a and b, The full-length AE3 in its inward-facing conformation model was generated by merging the TMD of hAE3^{HCO₃⁻} model and the NTDs/PIP₂ of the hAE3^{NTD₂TMD} model. Protomer A was shown as solvent-accessible electrostatic surface-potential maps, and protomer B was shown as a cartoon model. The pathogenic mutation sites were highlighted in yellow. The PIP₂ was shown as a stick model colored in pink.

P. 7 lines 19-21 – “capacity to lower acidity in response to significant alkalosis challenges in acid-secreting tissue” – this phrase is confusing.

Our response: We appreciate your suggestion, and we've revised “capacity to lower acidity” to “acid-loading effect” to make this phrase more comprehensible.

Lines 22-23 – what is meant by “In these environments [of excitable tissues], pH variations are more subtly tied to metabolic processes?” This phrase seems confused and unjustified – in fact it’s easier to image more extreme changes rather than gradual and moderate changes in excitable tissues.

Our response: Thank you to the reviewer for pointing out the confusing and unjustified phrase in our manuscript. To make this phrase more comprehensible, we have revised it as follows: "Maintaining pH homeostasis is especially important in these environments, as electrical activity can induce rapid pH changes. These acid-base fluctuations may impact physiological processes by affecting various ion channels^{13,14}. Thus, ensuring gradual and moderate pH changes in excitable tissues is crucial."

Line 30: change “carbonate” to “bicarbonate”

Our response: Thank you for the suggestion. We have changed “carbonate” to “bicarbonate”.

Lines 30-32: discussion of AE2 inner vestibular occupancy by C-terminal loop requires citation.

Our response: We thank the reviewer for the suggestion that a reference should be cited when discussing the inner vestibular occupancy of AE2 by the C-terminal loop. The appropriate reference has been cited as recommended.

Minor points:

P 2 Line 7: change “were evolved” to “evolved”

Our response: Thanks for the suggestion. We've replaced “were evolved” with “evolved”.

P2 Line 23: “transportation” should be “transport.”

Our response: Thank you for the suggestion. We have revised “transportation” to “transport”.

P 4 line 1 “adapts into” change to “adopts a homodimeric state”

Our response: We appreciate your suggestion, and we've revised “adapts into” to “adopts a homodimeric state”.

P 4 line 2: change “soluble” to “cytoplasmic”

Our response: Thank you for the suggestion. We have changed “soluble” to “cytoplasmic”

p. 22 line 16: change “stained” to “loaded”

Our response: Thanks for the suggestion. We've replaced "stained" with "loaded".

Reviewer #2 (Remarks to the Author)

In this manuscript by Jian L et al. the authors reveal the high resolution cryo-EM structure of human Anion Exchanger-3 (SLC4A3) bound to carbonate and the small molecule inhibitor DIDS. Revealing the structure of hAE3 is timely, as recent publications have also highlighted structures of hAE1 and hAE2. Thus, this manuscript provides one of the final missing pieces in the structural puzzle surrounding this family of anion exchangers. Moreover, the structures of hAE3 presented here provide structural context from which to interpret the different pH and DIDS sensitivities of transporters in this family.

In contrast to previous investigations with hAE1 and 2, the authors found that hAE3 predominately adopts the outward facing conformation, and attempts to obtain an inward facing structure of hAE3 were unsuccessful. Furthermore, the NTD soluble domains of hAE3 display significant flexibility in the outward facing structures of hAE3. To overcome these limitations the authors utilized previous inward facing structures to construct a (likely accurate) homology model of hAE3 in an inward facing conformation, and also constructed a hAE3(NTD)2(TMD) chimera to facilitate structural elucidation of the hAE3 flexible NTD soluble domain. The approaches utilized in this manuscript are novel, and provide critical insight into this important family of human Anion Exchangers.

Overall, the manuscript is well written, the figures are well designed and clear to aid the reader in interpretation, and all data both structural and functional appears to be of high quality. I have only limited points for the authors to consider before publication.

1. One part of the manuscript that I struggled with begins on line 167 (describing the anion binding pocket). I believe there are several errors in the numbering of residues in this paragraph. I believe the authors want to refer to residue K1173 throughout the paragraph. But this residue is misnumbered as K1133 (line 167) and K1153 (line 184). Please confirm that all numbering is consistent.

Our response: We thank you for highlighting the discrepancies in the residue numbering within the section describing the anion-binding pocket. We have corrected the misnumbered residues on lines 167 and 184.

2. On line 234 and 238, I believe the authors should point to supplemental figure 8B instead of 8A.

Our response: Thank you for pointing out the incorrect figure references in our manuscript. We have made the necessary corrections to lines 234 and 238.

3. Throughout the manuscript the authors refer to the cryo-EM map as "electron density". Cryo-EM maps are not electron density, they are a representative map of the coulomb potential of the protein (see ...Marques MA, Purdy MD, Yeager M. CryoEM maps are full of potential. Curr Opin Struct Biol. 2019. PMID: 31400843). I would recommend to avoid referring to cryo-EM maps as electron density.

Our response: We appreciate the reviewer's clarification of the distinction between "electron density" typically used in crystallography and the "coulomb potential map" represented in cryo-EM. In accordance with the suggestion, we have carefully revised the manuscript to replace all instances of "electron density" with "map density" to more accurately reflect the cryo-EM methodology.

4. In supplemental figure 6B it may be nice to show the map for the bound HCO₃⁻ if possible.

Our response: We thank the reviewer for the suggestion, and we have enhanced the Supplement Figure 6 by including the local map that specifically highlights the HCO₃⁻ ion along with the adjacent residues.

5. Figure 1E, the symmetry operator symbol showing the two-fold axis appears to be off centered.

Our response: Thank you for bringing this error to our notice. We have reviewed the figure and corrected the placement of the symbol to accurately represent the symmetry.

6. In Supplemental table 1 under the refinement section, the model resolution should be reported at a cutoff of 0.5 (not 0.143).

Our response: We thank the reviewer for the suggestion and have updated the table to reflect the model

resolution at an FSC cutoff of 0.5, as recommended.

Reviewer #3 (Remarks to the Author)

The manuscript by Jian and Cao et al reports structures of human anion exchanger 3 in the apo state, in the presence of a substrate, and in the presence of both the substrate and an inhibitor DIDS. Since the NTD of AE3 was not resolved the three structures, the authors determined the structure of a chimeric transporter with the NTD of AE3 and TMD of AE2. The structure of the chimera shows that the structure of AE3 NTD is similar to these of AE1 and AE3, and that the interactions between AE3 NTD to AE2 TMD are comparable to these of AE2 NTD and TMD. However, the strength of NTD/TMD interactions in AE3 could be weaker due to differences in amino acid sequences. The authors concluded that AE3 tends to stay in the outward facing conformation and that this preference is the cause for its higher sensitivity to DIDS and slower rate of transport and thus modest activity in pH regulation.

I feel that the structures are well done and these structures are part of our knowledge base for understanding of AE family of transporters. **I am not convinced that the lack of AE3 in the inward facing conformation on EM grids is sufficient evidence for its higher sensitivity to the inhibitor nor its slower rate of transport than the other AEs. If the authors are enthusiastic about these conclusions, then more precise experiments are needed, however, I should clarify that the comment is not a demand for more experiments.**

Our response: We thank the reviewer for the discussion on the relationship between the lack of AE3 in inward-facing conformation and its sensitivity to the inhibitor and transporting efficiency. In our analysis, we highlighted that the accessibility of DIDS to its binding site in the transmembrane domains (TMDs) appears predominantly when AE3 is in an outward-facing conformation. This observation was supported by our comparative studies with AE2, where we noted a greater challenge in forming the AE2-DIDS complex relative to the AE3-DIDS complex. Given that AE2 and AE3 exhibit different preferred conformations under the conditions used in our EM sample preparations, we suggested that AE3's propensity to adopt an outward-facing conformation might underlie its heightened response to DIDS in functional assays.

However, we acknowledge that this conclusion is drawn from cryo-EM observations and that further cellular experiments are indeed necessary to more definitively determine the conformational dynamics of AE2 and AE3 within the cell membrane. Based on the comments, we have revised the discussion section of our manuscript to clarify the speculative nature of our interpretations and to emphasize the need for additional experimental evidence.

The authors may want to change “electron density” to “density”. The former is reserved for X-ray diffraction methods.

Our response: We agree with the comments and have revised the manuscript accordingly.

References

1. Christiansen, M.K. et al. Genetic analysis identifies the SLC4A3 anion exchanger as a major gene for short QT syndrome. *Heart Rhythm* **20**, 1136–1143 (2023).
2. Salameh, A.I., Hübner, C.A. & Boron, W.F. Role of Cl⁽⁻⁾-HCO₃⁽⁻⁾ exchanger AE3 in intracellular pH homeostasis in cultured murine hippocampal neurons, and in crosstalk to adjacent astrocytes. *J Physiol* **595**, 93–124 (2017).
3. Hentschke, M. et al. Mice with a targeted disruption of the Cl⁽⁻⁾/HCO₃⁽⁻⁾ exchanger AE3 display a reduced seizure threshold. *Mol Cell Biol* **26**, 182–91 (2006).
4. Ko, S.B. et al. AE4 is a DIDS-sensitive Cl⁽⁻⁾/HCO₃⁽⁻⁾ exchanger in the basolateral membrane of the renal CCD and the SMG duct. *Am J Physiol Cell Physiol* **283**, C1206–18 (2002).
5. Parker, M.D. & Boron, W.F. The divergence, actions, roles, and relatives of sodium-coupled bicarbonate transporters. *Physiol Rev* **93**, 803–959 (2013).
6. Arakawa, T. et al. Crystal structure of the anion exchanger domain of human erythrocyte band 3. *Science* **350**, 680–4 (2015).
7. Capper, M.J. et al. Substrate binding and inhibition of the anion exchanger 1 transporter. *Nat Struct Mol Biol* **30**, 1495–1504 (2023).
8. Zhang, Q. et al. The structural basis of the pH-homeostasis mediated by the Cl⁽⁻⁾/HCO₃⁽⁻⁾ exchanger, AE2. *Nat Commun* **14**, 1812 (2023).
9. Vallese, F. et al. Architecture of the human erythrocyte ankyrin-1 complex. *Nat Struct Mol Biol* **29**, 706–718 (2022).
10. Zhang, W. et al. Structural and functional insights into the lipid regulation of human anion exchanger 2. *Nat Commun* **15**, 759 (2024).
11. Thorsen, K. et al. Loss-of-activity-mutation in the cardiac chloride-bicarbonate exchanger AE3 causes short QT syndrome. *Nat Commun* **8**, 1696 (2017).
12. Kovacs, B. et al. Two novel variants in the SLC4A3 gene in two families with Short QT Syndrome: the role of cascade screening. *European Heart Journal* **41** (2020).
13. Vaughan-Jones, R.D., Spitzer, K.W. & Swietach, P. Intracellular pH regulation in heart. *J Mol Cell Cardiol* **46**, 318–31 (2009).
14. Chesler, M. Regulation and modulation of pH in the brain. *Physiol Rev* **83**, 1183–221 (2003).

REVIEWERS' COMMENTS

Reviewer #1 (Remarks to the Author):

The authors have conscientiously responded to the reviewer critiques. Only very minor issues remain.

Lines 141-144. To facilitate reader understanding, discussion of the PGDKP sequence should be localized in the context of the transmembrane and linker helices schematized in Fig. 1D. Fig. 1f is mislabeled as Fig 1d.

Fig 6 legend. Should "color-coded by atomic element" rather be "by surface charge"?

Reviewer #2 (Remarks to the Author):

The authors have adequately addressed all concerns that were raised. I recommend publication of this manuscript.

Reviewer #3 (Remarks to the Author):

The authors have addressed my questions. I have no further comments.

Reviewer #1 (Remarks to the Author):

The authors have conscientiously responded to the reviewer critiques. Only very minor issues remain.

Lines 141-144. To facilitate reader understanding, discussion of the PGDKP sequence should be localized in the context of the transmembrane and linker helices schematized in Fig. 1D.

Our reply: We thank the reviewer for the suggestion. We have revised the paragraph as recommended, incorporating the discussion of the PGDKP sequence within the context of the transmembrane and linker helices. This adjustment aims to enhance reader understanding by providing a clearer structural context.

Fig. 1f is mislabeled as Fig 1d.

Our reply: We thank the reviewer for pointing out the potentially confusing label in the Figure legend. We have revised the caption accordingly.

Fig 6 legend. Should “color-coded by atomic element” rather be “by surface charge”?

Our reply: We thank the reviewer for the question. In the Fig. 6 legend, the phrase “color-coded by atomic element” specifically refers to residue R1052, which is depicted as a stick model colored by its atomic elements. Thank you for allowing us to clarify this detail.

Reviewer #2 (Remarks to the Author):

The authors have adequately addressed all concerns that were raised. I recommend publication of this manuscript.

Our reply: We thank the reviewer for the comments and suggestions throughout the review process, which have greatly improved the quality of our manuscript.

Reviewer #3 (Remarks to the Author):

The authors have addressed my questions. I have no further comments.

Our reply: We thank the reviewer for the comments and suggestions throughout the review process, which have greatly improved the quality of our manuscript.